# Mechanically stimulated ATP release from murine bone cells is regulated by a balance of injury and repair

**Nicholas Mikolajewicz[1,2], Elizabeth A Zimmermann[2,3], Bettina M Willie[2,3], Svetlana V Komarova[1,2]***

[1]Faculty of Dentistry, McGill University, Montreal, Quebec, Canada; [2]Shriners Hospital for Children - Canada, Montreal, Quebec, Canada; [3]Department of Pediatric Surgery, Montreal, Quebec, Canada

**Abstract** Bone cells sense and actively adapt to physical perturbations to prevent critical damage. ATP release is among the earliest cellular responses to mechanical stimulation. Mechanical stimulation of a single murine osteoblast led to the release of $70 \pm 24$ amole ATP, which stimulated calcium responses in neighboring cells. Osteoblasts contained ATP-rich vesicles that were released upon mechanical stimulation. Surprisingly, interventions that promoted vesicular release reduced ATP release, while inhibitors of vesicular release potentiated ATP release. Searching for an alternative ATP release route, we found that mechanical stresses induced reversible cell membrane injury *in vitro* and *in vivo*. $Ca^{2+}$/PLC/PKC-dependent vesicular exocytosis facilitated membrane repair, thereby minimizing cell injury and reducing ATP release. Priming cellular repair machinery prior to mechanical stimulation reduced subsequent membrane injury and ATP release, linking cellular mechanosensitivity to prior mechanical exposure. Thus, our findings position ATP release as an integrated readout of membrane injury and repair.
DOI: https://doi.org/10.7554/eLife.37812.001

*For correspondence:
svetlana.komarova@mcgill.ca

**Competing interests:** The authors declare that no competing interests exist.

## Introduction

The mechanical environment is an important determinant of bone health, as emphasized by a consistent bone loss in astronauts exposed to microgravity or in paralyzed or bedridden patients (*Nagaraja and Jo, 2014*). Gravitational and muscle forces act on the skeleton during physical activity resulting in a complex combination of shear forces, strains and pressures. Bone-embedded osteocytes and bone-forming osteoblasts are widely regarded as the mechanosensitive cells in the skeletal system (*Weinbaum et al., 1994*).

Following mechanical stimulation of rodent and human osteoblasts, transient intracellular free calcium ($[Ca^{2+}]_i$) elevations and ATP release are among the earliest detectable events, which result in autocrine and paracrine purinergic (P2) receptor signaling (*Robling and Turner, 2009*; *Romanello et al., 2001*; *Genetos et al., 2005*). Vesicular release of lysosomes or secretory vesicles, or conductive release via channels such as Maxi anion channels, Volume-regulated anion channels (VRAC), Connexins or Pannexins are the main mechanisms of regulated ATP release in mammalian cells (*Mikolajewicz et al., 2018*). Pathological ATP spillage also occurs from traumatically damaged cells (*Burnstock and Verkhratskii, 2012*). Of interest, non-lethal membrane injury has been demonstrated *in vivo* in several tissues (*McNeil and Steinhardt, 2003*), including bone (*Yu et al., 2017*), under physiological conditions. The mechanism of facilitated cell membrane repair has been described and involves $Ca^{2+}$/PKC-dependent vesicular exocytosis (*Togo et al., 1999*). However, the contribution of non-lethal cell injury to ATP release and related mechanotransductive purinergic signaling remains unclear.

**eLife digest** Athletes' skeletons get stronger with training, while bones weaken in people who cannot move or in astronauts experiencing weightlessness. This is because bone cells thrive when exposed to forces. When a bone cell is exposed to a physical force, the first thing that happens is the release of the energy-rich molecule called ATP into the space outside the cell. This molecule then binds to the neighboring cell to unleash a cascade of responses. ATP can exit the cell either through special canals in the cell membrane or released in tiny pod-like structures called vesicles. It is known that strong forces can injure the cell membrane and cause ATP to spill out. However, it is less clear how ATP is released when cells are subjected to regular forces.

Mikolajewicz et al. investigated whether ATP exits through injured membranes of cells experiencing regular forces. Bone cells grown in the laboratory were gently poked with a glass needle or placed in a turbulent fluid to simulate forces experienced in the body. Dyes and fluorescent imaging techniques were used to observe the movement of vesicles and calculate the concentration of ATP in these cells.

The experiments showed that regular forces in the body do indeed injure the cell membranes and cause ATP to spill out. But importantly, the cells repaired the injuries quickly by releasing vesicles that patch the wound. As soon as the membrane is sealed, ATP stops coming out. From the first injury, cells adapted and quickly strengthened their membrane and repair system to be more resilient against future forces. This process was also seen in the shin bones of mice.

These results are important because knowing how bone cells sense, respond and convert physical forces can help us develop treatments for astronauts, the injured and aged.
DOI: https://doi.org/10.7554/eLife.37812.002

The goal of this study was to examine the mechanism of ATP release from mechanically stimulated cells of the osteoblastic lineage. Since we have previously demonstrated that transient membrane disruption is required to induce global $[Ca^{2+}]_i$ elevations in osteoblasts (*Lopez-Ayon et al., 2014*), we were particularly interested in understanding the contribution of membrane injury to mechanically induced ATP release. Mechanical forces were applied by local membrane deformation or turbulent fluid shear stress *in vitro* to BMP-2 transfected C2C12 osteo-blastic cells (C2-OB), primary bone marrow (BM-OB) and compact bone (CB-OB)-derived osteoblasts and changes in $[Ca^{2+}]_i$, vesicular exocytosis, membrane permeability and ATP release were assessed. The prevalence of membrane injury in osteocytes at physiological and supraphysiological mechanical strain levels was investigated following *in vivo* cyclic compressive tibial loading of 10-week-old female C57Bl/6J mice.

## Results

### Mechanically stimulated osteoblasts release ATP that induces calcium responses in non-stimulated neighboring cells

Osteoblasts from three different sources, C2-OB, CB-OB, and BM-OB, were loaded with $[Ca^{2+}]_i$ dye Fura2 and mechanically stimulated with a glass micropipette, which induced qualitatively similar transient global $[Ca^{2+}]_i$ elevations, consistent with prior work (*Robling and Turner, 2009*; *Romanello et al., 2001*; *Genetos et al., 2005*) (*Figure 1A–C, Figure 1—video 1*). L-type voltage-sensitive calcium channel (VSCC) inhibitor Nifedipine and P2 antagonist PPADS significantly reduced the amplitude of mechanically-stimulated $[Ca^{2+}]_i$ transients (*Figure 1D*). L-type VSCC activation occurred gradually (*Figure 1E*) while the P2 receptor-driven component of the response peaked within seconds of stimulation (*Figure 1F*). Together, L-type VSCC and P2 receptor-driven compo-nent accounted for ~50% of the mechanical stimulated $[Ca^{2+}]_i$ transient. Consistent with previous reports (*Robling and Turner, 2009*; *Romanello et al., 2001*; *Genetos et al., 2005*), shortly after a single osteoblast was mechanically stimulated, neighboring cells exhibited delayed secondary $[Ca^{2+}]_i$ responses (*Figure 1G*). Pharmacological interventions revealed that P2 receptors mediated the sec-ondary response in all three osteoblast models, while a tendency for GAP junction involvement was

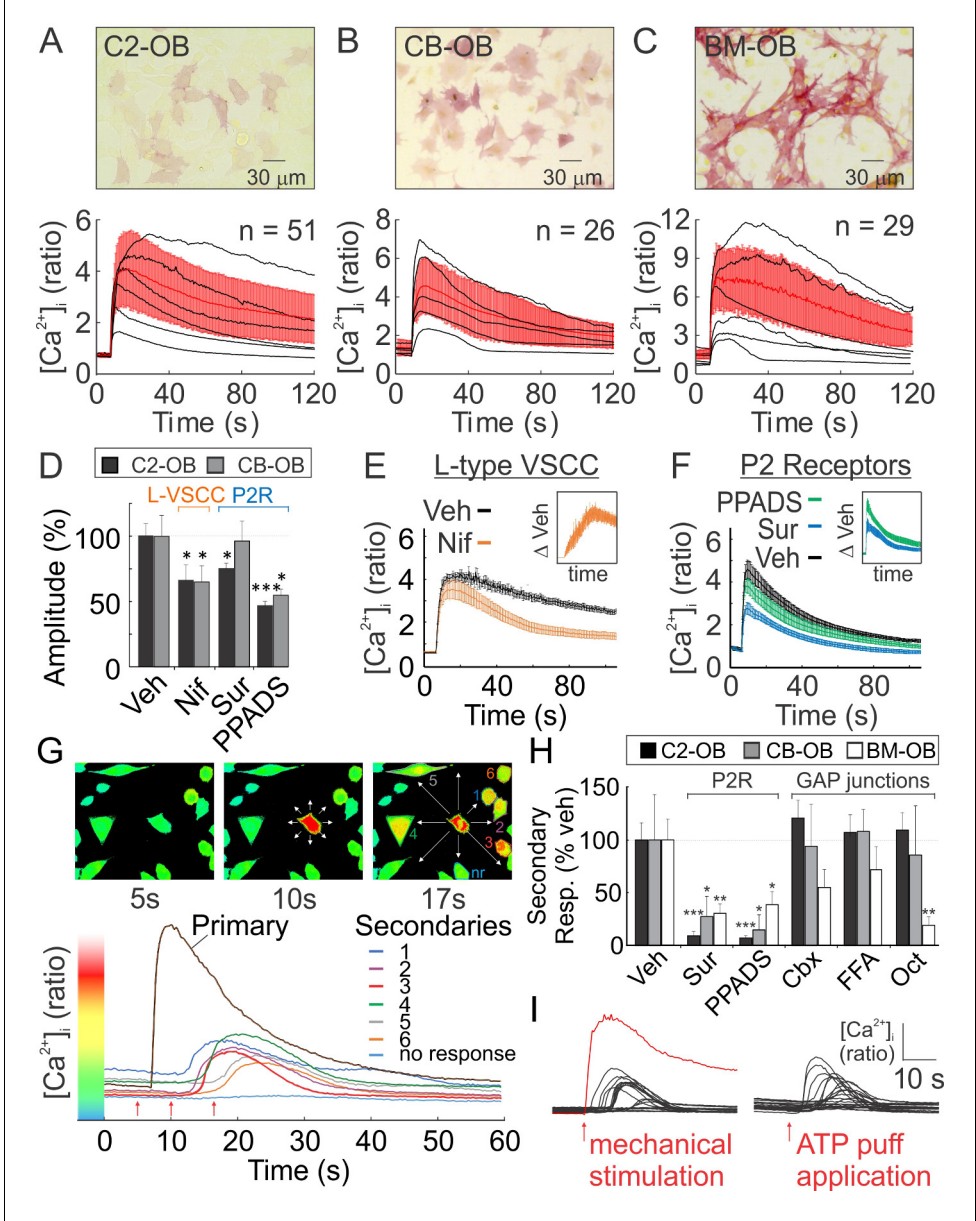

**Figure 1.** Osteoblasts are mechanosensitive (A-C) Single Fura2-loaded C2-OB (A), CB-OB (B) or BM-OB (C) (*Top*: ALP staining) were stimulated with micropipette and $[Ca^{2+}]_i$ was recorded (*Figure 1—video 1*). *Bottom*: representative traces (*black*), means ± SEM (*red*) (D) Amplitudes of mechanically evoked $[Ca^{2+}]_i$ transients in osteoblasts pretreated with vehicle, Nif, Sur or PPADS (see Materials and methods for abbreviations, targets and doses). Means ± SEM, n = 6–15 stimulated cells, normalized to vehicle. (E, F) Contribution of L-type VSCC (E, n = 10) and P2 receptors (F, n = 12) to mechanically evoked $[Ca^{2+}]_i$ transients in C2-OB as comparison and difference (*insets*) of mean ±SEM of vehicle and treatment $[Ca^{2+}]_i$ transients. (G) $[Ca^{2+}]_i$ in the stimulated (primary) and neighboring (secondary) C2-OB. *Top*: pseudocolor of 340/380 ratio, *white arrows*: directions of signal propagation, *bottom*: $[Ca^{2+}]_i$ transients, *red arrows*: top panel time points. (H) Secondary responsiveness in cultures pretreated with vehicle, Sur, PPADS, Cbx, FFA or Oct. Means ± SEM, n = 7–21 stimulated primary responses, normalized to vehicle. (H) $[Ca^{2+}]_i$ transients induced by mechanical stimulation of single C2-OB (*left*, red curve) or puff application of 10 µM ATP (*right*). For *Figure 1*, *significance between treatment and control by ANOVA. Source data for *Figure 1* is provided in *Figure 1—source data 1*.

DOI: https://doi.org/10.7554/eLife.37812.003

The following video and source data are available for figure 1:

**Source data 1.**

DOI: https://doi.org/10.7554/eLife.37812.005

**Figure 1—video 1.** Single Fura2-loaded C2-OB was stimulated with micropipette and $[Ca^{2+}]_i$ was recorded.

DOI: https://doi.org/10.7554/eLife.37812.004

observed in BM-OB responses (*Figure 1H*). Puff application of 10 μM ATP mimicked the appearance of secondary responders in C2-OB (*Figure 1I*).

We used a luciferin fluorescence-based imaging technique (*Sørensen and Novak, 2001*) to directly measure ATP release ([ATP]$_e$) (*Figure 2—figure supplement 1*). Simultaneous recording of [ATP]$_e$ and [Ca$^{2+}$]$_i$ demonstrated that ATP release occurred within seconds after mechanical stimulation (time to peak release: 1.64 ± 0.36 s) and preceded the onset of [Ca$^{2+}$]$_i$ responses in neighboring cells (*Figure 2*). We measured pericellular [ATP]$_e$ concentrations in the range of 0.05–80.5 μM corresponding to a release of 70 ± 24 amole ATP from a single mechanically stimulated osteoblast (*Figure 2B*). Pericellular [ATP]$_e$ significantly correlated with the percentage of secondary responders (*Figure 2C*).

## Vesicular ATP released upon mechanical stimulation is not the primary source of extracellular ATP

Vesicular ATP release was proposed to be a major source of extracellular ATP (*5*). Using confocal microscopy, we found that an acidophilic marker quinacrine and a fluorescent ATP analog MANT-ATP co-localized in intracellular granular compartments (*Figure 3A*; *Figure 3—video 1–2*). Quinacrine-positive vesicular pool was entirely released after treatment with Ca$^{2+}$ ionophore ionomycin, and only partially by mechanical stimulation (*Figure 3B*; *Figure 3—video 3–5*), while negligible vesicular release was observed in unstimulated cells. Interestingly, vesicular release peaked within 20 s of mechanical stimulation (*Figure 3C, D*), while ATP release within ~2 s (*Figure 2A*). Basal vesicular density was 70 ± 4×10$^{-3}$ vesicles/μm$^2$ (*Figure 3E*) and mechanical stimulation resulted in the

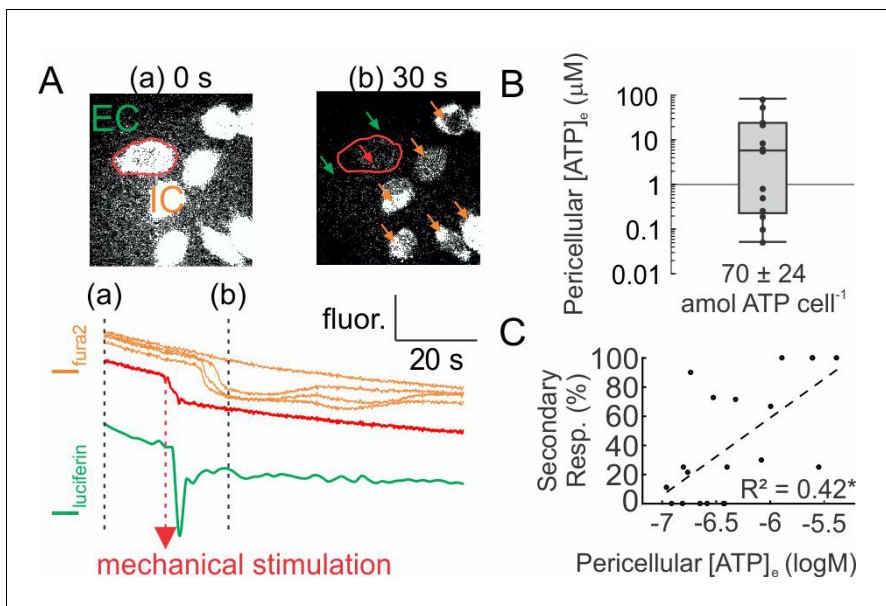

**Figure 2.** ATP release following mechanical stimulation of a single osteoblast. (**A**) Fura2-loaded C2-OB bathed in luciferin/luciferase containing PS was micropipette-stimulated. *Top*: 380 ex/510 em images before (**a**) and after (**b**) stimulation of the cell (*red outline*). *Bottom*: intracellular (IC) Fura2 fluorescence [I$_{fura2}$] of primary (*red*) and secondary (*orange*) responders and extracellular (EC) luciferin fluorescence [I$_{luciferin}$] (*green*). (**B**) Box plot of [ATP]$_e$ in pericellular region of stimulated C2-OB, *black markers*: independent observations, n = 13 stimulated cells. (**C**) Correlation between [ATP]$_e$ and percentage of secondary responders observed in C2-OB (n = 20 stimulated primary responses), *dashed line*: linear regression. Source data for *Figure 2* is provided in *Figure 2—source data 1*.

DOI: https://doi.org/10.7554/eLife.37812.006

The following source data and figure supplement are available for figure 2:

**Source data 1.**
DOI: https://doi.org/10.7554/eLife.37812.008
**Figure supplement 1.** Real-time ATP measurement assay using fluorescent properties of luciferin-luciferase.
DOI: https://doi.org/10.7554/eLife.37812.007

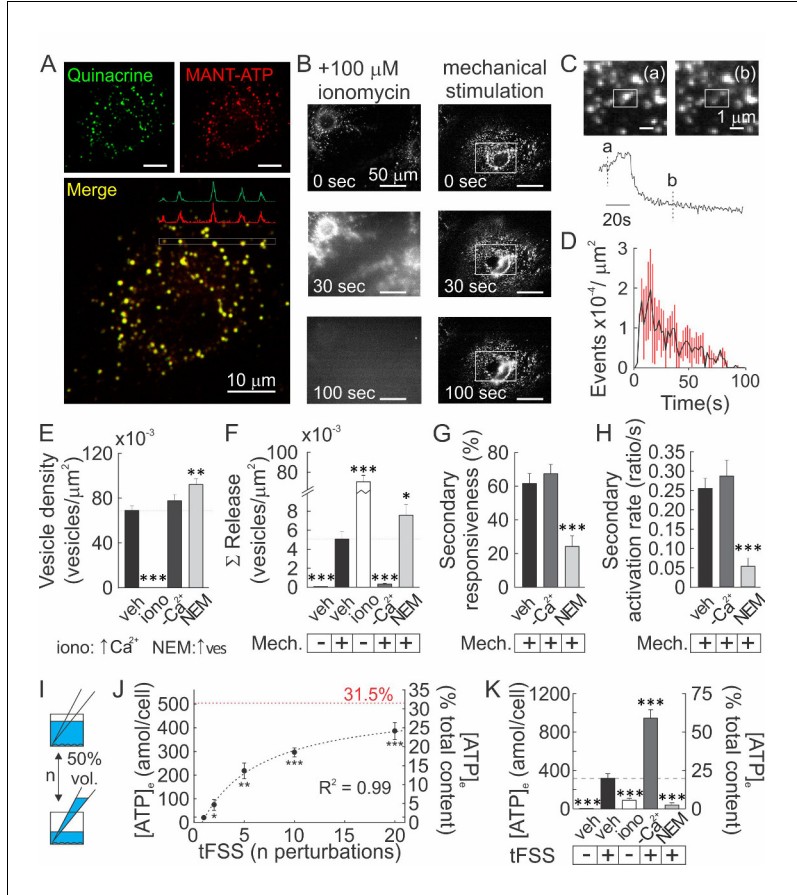

**Figure 3.** Vesicular ATP released upon mechanical stimulation is not the primary source of extracellular ATP. (A) Confocal images of CB-OB loaded with quinacrine and MANT-ATP, *Figure 3—video 1-2*. (B) Changes in quinacrine-loaded CB-OB treated with ionomycin (*left*) or stimulated with micropipette (*right*, region of interest in white). (C) Vesicular release event in CB-OB (*top*) coincides with a sudden drop in granular quinacrine fluorescence (*bottom*), *Figure 3—video 3-5*.(D) Kinetics of vesicular release from CB-OB stimulated by micropipette at 0 s, n = 37 stimulated cells. (E–H) Quinacrine- (E, F) or Fura2- (G, H) loaded CB-OB were pre-treated (10 min) with vehicle, ionomycin (iono) or NEM or placed in $[Ca^{2+}]$-depleted physiological solution, and micropipette-stimulated when indicated (+). Vesicular density (E) and cumulative release (F), secondary responsiveness (G) and $[Ca^{2+}]$ response activation rates (H) were determined, n = 7–37 primary cells. (I) tFSS was applied by replacing 50% media volume *n* times. (J, K) ATP released per cell (*left axis*) or as percent of cellular ATP content (*right axis*) was measured following CB-OB stimulation by tFSS (J, *black dashed line*: rational function fit; *red dashed line:* corresponding asymptote) or after indicated pre-treatments followed by tFSS (K, 10x media displacements, +), n = 6–8 independent cultures. For *Figure 3*, means ± SEM, *significance compared to vehicle (E–H), basal ATP release (J) or to tFSS-stimulated vehicle (K) by ANOVA. Source data for *Figure 3* is provided in *Figure 3—source data 1*.
DOI: https://doi.org/10.7554/eLife.37812.009

The following video, source data, and figure supplement are available for figure 3:

**Source data 1.**
DOI: https://doi.org/10.7554/eLife.37812.017
**Figure supplement 1.** Involvement of conductive channels in osteoblast response to mechanical stimulation.
DOI: https://doi.org/10.7554/eLife.37812.010
**Figure 3—video 1.** Confocal Z-stack of quinacrine- (left) and MANT-ATP (middle)-loaded CB-OBcells with merged images (right), ex. 1.
DOI: https://doi.org/10.7554/eLife.37812.011
**Figure 3—video 2.** Confocal Z-stack of quinacrine- (*left*) and MANT-ATP (*middle*)-loaded CB-OB cells with merged images (*right*), ex. 2.
DOI: https://doi.org/10.7554/eLife.37812.012
**Figure 3-—video 3.** Single quinacrine-loaded CB-OB was stimulated with micropipette and vesicular release events were recorded, ex. 1.
DOI: https://doi.org/10.7554/eLife.37812.014
**Figure 3—video 4.** Single quinacrine-loaded CB-OB was stimulated with micropipette and vesicular release events were recorded, ex. 2.
DOI: https://doi.org/10.7554/eLife.37812.015
**Figure 3—video 5.** Single quinacrine-loaded CB-OB was stimulated with micropipette and vesicular release events were recorded, ex. 3.
DOI: https://doi.org/10.7554/eLife.37812.016

cumulative release of $5 \pm 1 \times 10^{-3}$ vesicles/µm$^2$ over 100 s post-stimulation (7.2 ± 1.2% of the basal vesicular pool) (*Figure 3F*). In [Ca$^{2+}$]$_e$-free conditions vesicular density was unaffected, whereas mechanically evoked vesicular release was markedly suppressed (*Figure 3E, F*). Treatment with N-ethylmaleimide (NEM) caused a 33 ± 2% increase in the vesicular density (*Figure 3E*), consistent with a NEM-induced increase in the readily releasable vesicular pool (*Lonart and Südhof, 2000*), and a 55 ± 8% increase in mechanically-evoked vesicular exocytosis (*Figure 3F*). Surprisingly, the percentage of secondary responders and activation rates for Ca$^{2+}$ responses (response amplitude divided by time for signal to increase from 10 to 90% of peak) were unaffected in [Ca$^{2+}$]$_e$-free conditions, whereas NEM induced a significant decline in the percentage of secondary responders and activation rates (*Figure 3G, H*). We measured bulk ATP release following mechanical stimulation by media displacement, a model for turbulent fluid shear stress (tFSS, *Rumney et al., 2012*, *Figure 3I*). Proportional to the extent of tFSS, 21 ± 11 to 422 ± 97 amol ATP/cell were released, approaching 31.5% of total intracellular ATP at high tFSS (*Figure 3J*). tFSS (10x media displacements) induced release of 324 ± 50 amol ATP/cell, while ionomycin only led to release of 94 ± 5 amol ATP/cell. Moreover, tFSS-induced ATP release was significantly potentiated under [Ca$^{2+}$]$_e$-free conditions (961 ± 88 amol ATP/cells) and inhibited in NEM-treated cells (41 ± 24 amol ATP/cells) (*Figure 3K*). We also investigated conductive channels, however, pharmacological inhibition of Maxi-anion channels, P2X7, Connexins, Pannexins, T-type VSCC, TRPV4 or Piezo1 channels did not affect tFSS-stimulated ATP release (*Figure 3—figure supplement 1*). Although inhibition of L-Type VSCC had a small inhibitory effect (*Figure 3—figure supplement 1*), our data suggest that conductive channels cannot account for ATP release observed from mechanically stimulated osteoblasts. Importantly, our data show that increases in vesicular release were consistently associated with a reduction in mechanically stimulated ATP release, while inhibition of vesicular exocytosis increased ATP release from mechanically-stimulated osteoblasts.

## Mechanical stimuli regularly and reversibly compromise the integrity of the osteoblast membrane

We hypothesized that mechanically stimulated ATP release is related to membrane injury. Adopting a Fura2-based dye-leakage assay (*Togo et al., 1999*), we examined changes in 340 ex/510 em Fura2 fluorescence following mechanical stimulation of osteoblasts (*Figure 4A*). In ~40% of micropipette-stimulated osteoblasts, intracellular Fura2 fluorescence returned to pre-stimulation baseline (minor or no injury; mIn). However, in 30–40% of osteoblasts post-stimulation fluorescence was partially reduced (intermediate injury; iIn) and in 15–20% of osteoblasts completely lost (severe injury; sIn) (*Figure 4B*). The extent of injury sustained by a micropipette-stimulated cell correlated with the magnitude and duration of [Ca$^{2+}$]$_i$ elevations (*Figure 4C–E*), as well as the percentage of secondary responders (*Figure 4F*), suggesting that ATP release is related to the extent of membrane disruption. In a dye uptake assay, we added membrane impermeable dyes, trypan blue (TB) or rhodamine-conjugate dextran (R-dextran), to osteoblasts before or after mechanical stimulation (*Figure 4G*). When the dyes were applied prior to tFSS, the proportion of dye-permeable osteoblasts positively correlated with the degree of tFSS. The extent of dye uptake was higher for TB (~900 Da) compared to R-dextran (~10 kDa) (*Figure 4H,I*). However, when membrane permeability was examined 300 s after stimulation, a significantly lower percentage of cells were permeable to the dyes (*Figure 4H,I*). Hemichannel involvement was excluded, since hemichannel blockers did not affect mechanically-stimulated [Ca$^{2+}$]$_i$ elevations, membrane permeability or ATP release (*Figure 4—figure supplement 1*). Consistent with transient membrane disruption, lactate dehydrogenase (LDH, ~140 kDa) leaked into the extracellular media proportionally to tFSS magnitude (*Figure 4J*), even though cell viability was minimally affected 1 hr after tFSS stimulation (*Figure 4K*). Examining TB uptake following micropipette stimulation, we have found that all C2-OB were TB-permeable immediately upon stimulation, decreasing to 75 ± 22% within 10 s, to 40–50% by 180 s and to 18 ± 12% by 300 s (*Figure 4L*). Comparing these data to dye leakage experiments suggested that minor injuries resealed within 10 s, intermediate injuries within 20–180 s and severe injuries did not reseal by 300 s. (*Figure 4L*). Based on predicted molecular radii of the dyes (*Erickson, 2009*), we estimated the membrane lesion radius to be on the nanometer scale, sufficiently large to permit the efflux of smaller ATP molecules (~507 Da) (*Figure 4M*).

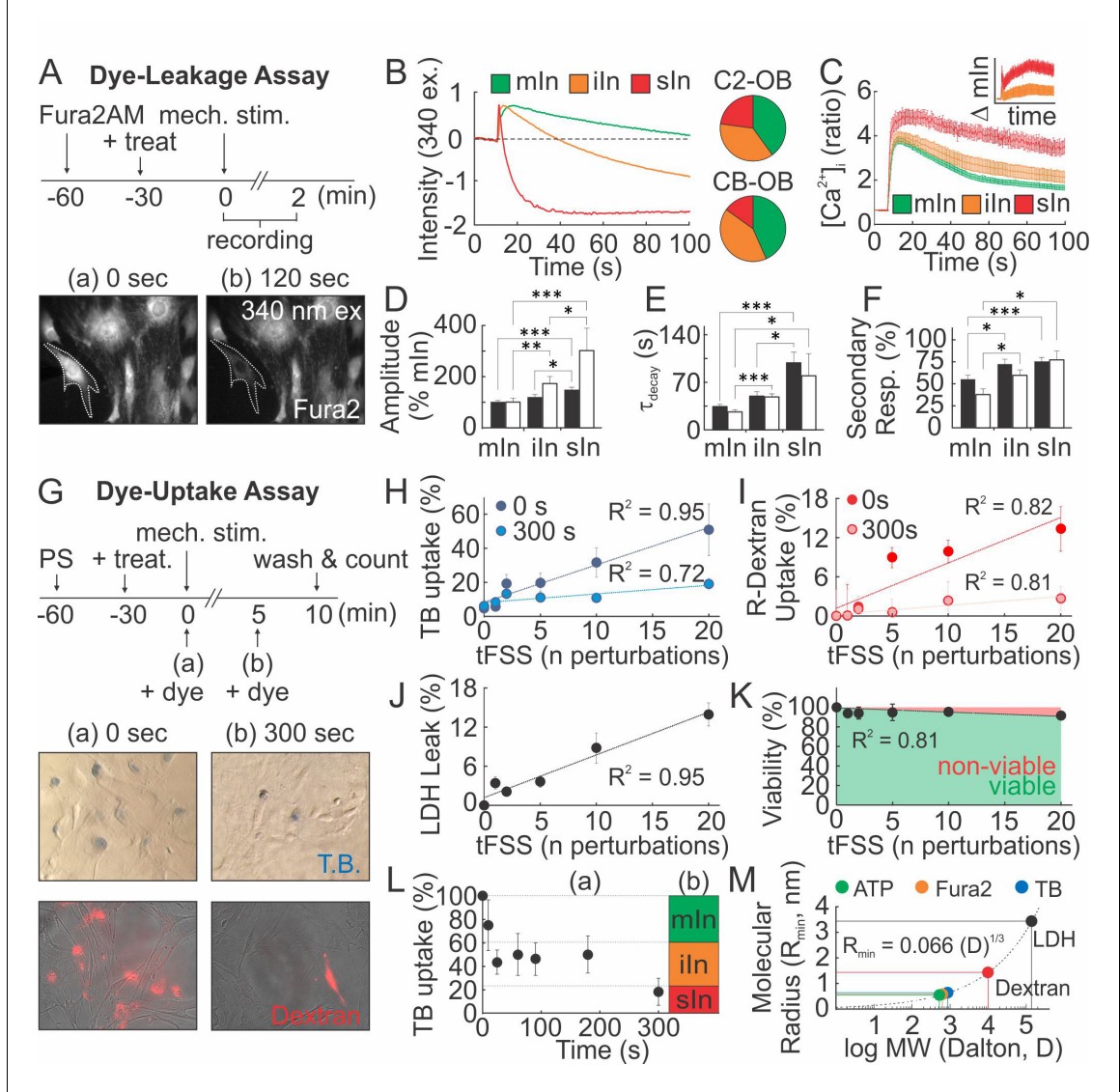

**Figure 4.** Mechanical stimuli regularly and reversibly compromise the integrity of osteoblast cell membrane. (A–F) Membrane permeability in micropipette-stimulated Fura2-loaded osteoblasts. (A) *Top:* Schematic of dye-leakage assay. *Bottom*: 340 ex/510 em before (a) and after (b) CB-OB stimulation (*white outline*). (B) Representative Fura2 traces in C2-OB with minor (mln), intermediate (iln) or severe (sln) cell injury and relative occurrence frequency (*right*, n = 35–40 stimulated cells). (C) C2-OB $[Ca^{2+}]_i$ responses and differences (*inset*) for mln (*green*), iln (*orange*) and sln (*red*), n = 8–14 stimulated cells. (D–F) Primary $[Ca^{2+}]_i$ response amplitudes (D) and decay constants (E), and secondary responsiveness (F) in CB-OB (*black*; n = 40 stimulated cells) or C2-OB (*white*; n = 35 stimulated cells), grouped by cell injury status. (G–K) Membrane permeability in tFSS-stimulated osteoblasts. (G) *Top:* schematic of dye-uptake assay. *Bottom*: CB-OB stained with TB (*top*) or R-dextran (*bottom*) prior to (0 s, *left*) or after (300 s, *right*) tFSS application (10x). Uptake of TB (H) or R-dextran (I) added prior to (0 s) or after (300 s) tFSS stimulation of CB-OB (n = 4 independent cultures). (J) Leakage of LDH from CB-OB 5 min after tFSS (n = 8 independent cultures, normalized to total LDH). (K) Viability of C2-OB 1 hr after tFSS assessed by alamarBlue (n = 8–16 independent cultures). (L) Uptake of TB added at indicated times after micropipette-stimulation of C2-OB (L-a, n = 4–23 stimulated cells) compared to relative frequencies of mln, iln and sln assessed by Fura2-leakage assay (L-b). (M) Calculated minimum membrane lesion radius $R_{min}$ required for permeability to LDH, R-dextran, TB, Fura2 and ATP. For *Figure 4*, means ± SEM, *dashed lines*: linear regression, *significance by ANOVA. Source data for *Figure 4* is provided in *Figure 4—source data 1*.

DOI: https://doi.org/10.7554/eLife.37812.018

The following source data and figure supplement are available for figure 4:

**Source data 1.**
DOI: https://doi.org/10.7554/eLife.37812.020
**Figure supplement 1.** Involvement of hemichannels in ATP release and membrane resealing in murine osteoblasts
DOI: https://doi.org/10.7554/eLife.37812.019

## Osteocyte membranes are reversibly disrupted during mouse *in vivo* tibia loading

To determine whether mechanically induced repairable membrane disruptions occur *in vivo*, 10-week-old female C57Bl/6J mice were injected with lysine-fixable Texas Red-conjugated dextran (LFTR-Dex, ~10 kDa) either 30 min prior to or 20 min after cyclic compressive tibial loading to strain magnitudes at the upper level of physiological activities (600 με) or at supraphysiological osteogenic levels (1200 με) (*Willie et al., 2013*) (*Figure 5A–D*, *Figure 5-video 1*). In the contralateral non-

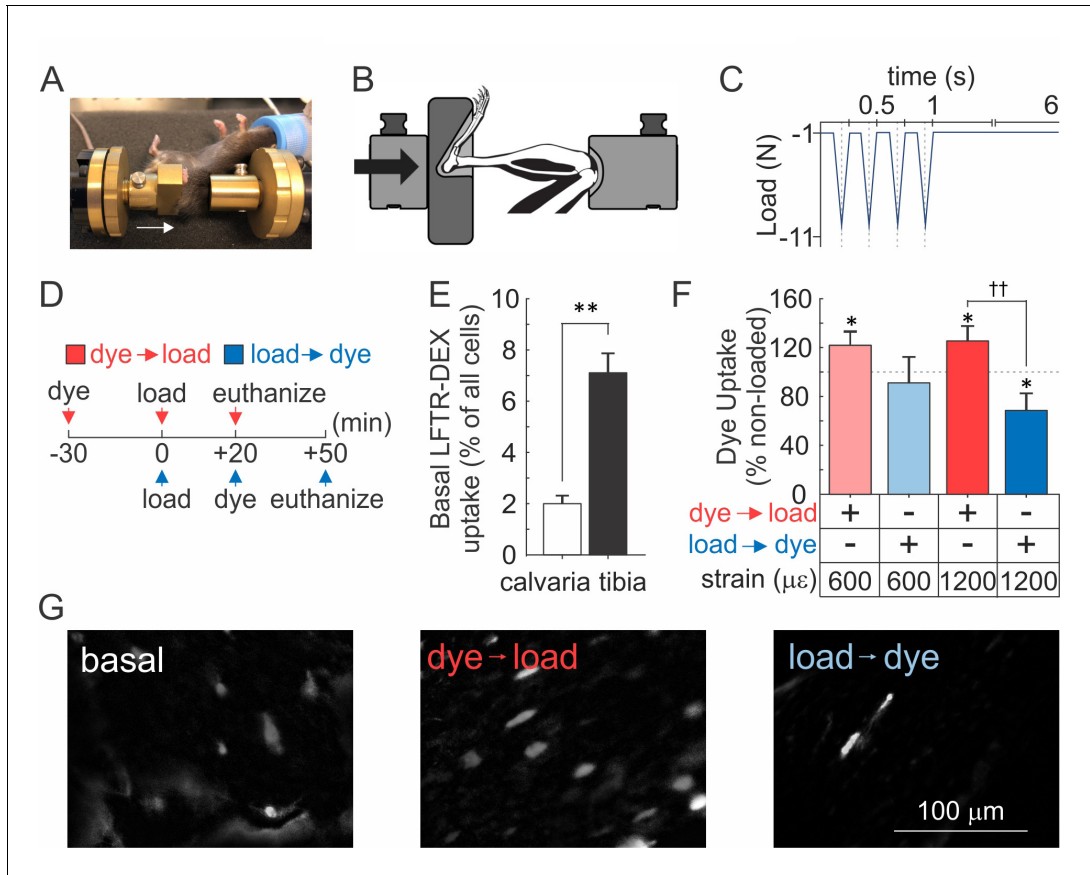

**Figure 5.** Osteocyte membranes are reversibly disrupted during *in vivo* cyclic compressive loading of the tibia. (A, B) Left tibia of anesthetized mouse was positioned in loading device as shown in picture (**A**) and schematic (**B**). *Arrows* indicate direction of load. (**C**) The triangle waveform included 0.15 s symmetric active loading/unloading, with a 0.1 s rest phase (−1 N) between load cycles and a 5 s rest inserted between every four cycles. A maximum force of −5.5 N or −11 N was applied, which engenders 600 με or 1200 με, respectively at the periosteal surface of the tibia mid-diaphysis in these mice. (**D**) Experimental design schematic: animals were injected with LFTR-Dex 30 min before (*red*) or 20 min after (*blue*) *in vivo* cyclic tibial loading (5 min). (**E**) Proportion of cells exhibiting LFTR-Dex uptake in calvariae (n = 5 animals) and control tibiae (n = 20 from 10 animals), normalized to average total cell number, means ± SEM, **p<0.01. (**F**) Dye uptake in tibia injected before (*red*) or after (*blue*) loading at 600 με or 1200 με, means ± SEM, n = 5/ strain/dye protocol, normalized to non-loaded contralateral tibia, *significance compared to non-loaded tibia, ††significance between dye protocols, by ANOVA. (**G**) Contrast enhanced images of dye uptake in control (*left*), and 1200 με loaded tibia injected with dye before (*middle*) or after (*right*) loading. Source data for *Figure 5* is provided in *Figure 5—source data 1*.

DOI: https://doi.org/10.7554/eLife.37812.021

The following video, source data, and figure supplement are available for figure 5:

**Source data 1.**
DOI: https://doi.org/10.7554/eLife.37812.023
**Figure supplement 1.** *In vivo* cyclic compressive loading was applied to the left tibia.
DOI: https://doi.org/10.7554/eLife.37812.022
**Figure 5—video 1.** Recording of *in vivo* cyclic compressive loading being applied to left tibia of anesthetized C57Bl/6J mouse.
DOI: https://doi.org/10.7554/eLife.37812.026

loaded control tibiae (which experienced habitual loads only), sclerostin-positive osteocytes were observed at a density of 3212 ± 287 cells/mm$^2$ (*Figure 5—figure supplement 1*),~4–7% of which demonstrated LFTR-Dex uptake, while only 2% of calvarial osteocytes were LFTR-Dex positive (*Figure 5E*). Cyclic compressive tibial loading in the presence of LFTR-Dex resulted in significantly increased osteocyte dye uptake compared to unloaded bone, demonstrating that *in vivo* mechanical loading results in cellular membrane disruption (*Figure 5F,G*, *red*). Importantly, when LFTR-Dex was administered 20–50 min after 600 με loading, the levels of cellular dye uptakes were similar to those in unloaded tibia, suggesting that damage evident immediately after loading was repaired within this time period (*Figure 5F, light blue*). Of interest, when LFTR-Dex was administered 20–50 min after 1200 με loading, the proportion of osteocyte exhibiting dye uptake was significantly lower compared to non-loaded controls, suggesting that adaptive improvements in membrane integrity occurred in response to the supraphysiological strain (*Figure 5F,G*, *dark blue*).

## PKC regulates vesicular exocytosis and membrane resealing

Vesicular exocytosis has been demonstrated to facilitate membrane repair and improve membrane integrity in a PKC-dependent manner (*Togo et al., 1999*). We examined the role of Ca$^{2+}$/PLC/PKC pathway during mechanically induced membrane injury and resealing. Pharmacological activation of PKC (*Figure 6A*, *Figure 6—figure supplement 1A*) and NEM-induced potentiation of vesicular release (*Figure 6—figure supplement 2A*) both attenuated tFSS-induced membrane disruption. Depletion of [Ca$^{2+}$]$_e$, or pharmacological inhibition of PLC (*Figure 6—figure supplement 2A,B*) or PKC (*Figure 6A,B*; *Figure 6—figure supplement 1A,B*) increased membrane disruption in osteoblasts following tFSS or micropipette stimulation. Together these findings suggest that membrane repair was regulated by extracellular calcium influx, PLC/PKC signaling and vesicular exocytosis. We next investigated the link between Ca$^{2+}$/PLC/PKC pathway and mechanically stimulated [Ca$^{2+}$]$_i$ elevations, vesicular exocytosis and ATP release. Depletion of [Ca$^{2+}$]$_e$ or PLC inhibition consistently reduced the amplitude of mechanically-stimulated [Ca$^{2+}$]$_i$ elevations (*Figure 6—figure supplement 2C*) and vesicular release (*Figure 6—figure supplement 2D*), and potentiated ATP release (*Figure 6—figure supplement 2E*). PKC activation had minor effect on calcium responses (*Figure 6—figure supplement 2C*), augmented vesicular release (*Figure 6C*) and reduced ATP release (*Figure 6D*, *Figure 6—figure supplement 1C*). In contrast, PKC inhibition potentiated calcium responses (*Figure 6—figure supplement 2C*), suppressed vesicular release (*Figure 6C*) and increased ATP release (*Figure 6D*, *Figure 6—figure supplement 1C*). PMA-induced PKC activation rescued vesicle pool depletion following PLC inhibition, thereby positioning PKC signaling downstream of PLC (*Figure 6—figure supplement 3*). Together these data suggest that mechanically stimulated membrane injury and repair, vesicular release and ATP release are regulated by Ca$^{2+}$/PLC/PKC-dependent signaling.

We next examined which PKC isoform is involved in the mechanoresponse. Immunoblot analysis demonstrated that mechanical stimulation (using tFSS) of CB-OB activated conventional PKCα/β and PKD/PKCμ, but not atypical PKCζ/λ or novel PKCδ/θ isoforms (*Figure 6E*, *Figure 6—figure supplement 4*). PMA stimulated phosphorylation of conventional and PKD/PKCμ isoforms, but not atypical or novel PKC isoforms, while only basal PKD/PKCμ phosphorylation had a tendency (p=0.09) to be inhibited by Bis. Total PKC levels were unaffected by pharmacological interventions or mechanical stimulation (*Figure 6—figure supplement 4G*). These findings suggest that the PKD/PKCμ isoform regulates membrane resealing in osteoblasts.

To establish the relationship between vesicular release, membrane disruption and ATP release, we pooled data for the mechanically induced responses from all treatments studied in CB-OB (*Figure 6F–H*). Mechanically induced vesicular release negatively correlated with the extent of membrane disruption (*Figure 6F*) and the amount of ATP released (*Figure 6G*), while membrane disruption positively correlated with amount of ATP released in response to mechanical stimulation (*Figure 6H*). These data support a model in which mechanical disruption of the cell membrane leads to ATP spillage, while the extent of membrane resealing, regulated by Ca$^{2+}$/PLC/PKC-dependent vesicular exocytosis, limits the amount of total ATP released (*Figure 7*).

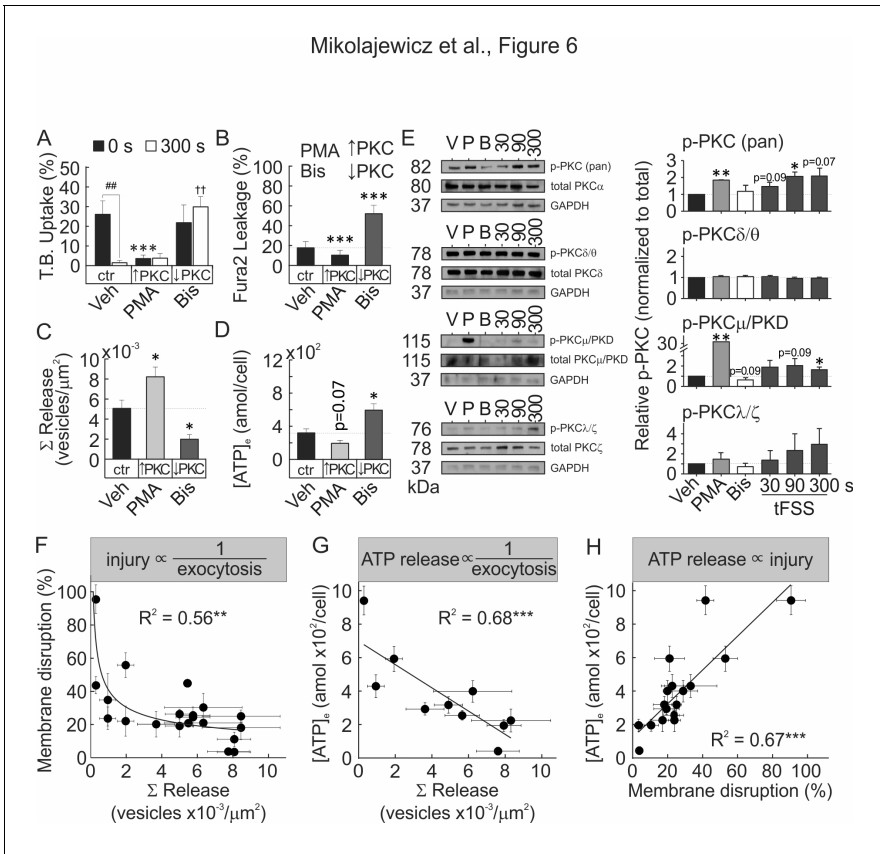

**Figure 6.** Membrane resealing depends on PKC-regulated vesicular exocytosis. (A–D) CB-OB were pretreated with vehicle, PMA or Bis, and membrane injury (A, B) vesicular exocytosis (C) and ATP release (D) were assessed. (A) TB uptake at 0 and 300 s after tFSS (10x), n = 5–8 independent cultures. (B) Fura2 leakage, determined as percentage of sIn cells after micropipette-stimulation, n = 9–16 stimulated cells. (C) Cumulative vesicular release over 100 s after micropipette-stimulation, n = 6–10 stimulated cells. (D) ATP release over 60 s following 10x tFSS stimulation, n = 8 independent cultures. (E) Immunoblotting for PKC isoform phosphorylation in CB-OB treated with vehicle, PMA or Bis (30 min); or 30 s, 90 s or 300 s after tFSS (10x). Shown are immunoblots (*left*), complete gels in *Figure 6—figure supplement 4*, and densitometry (*right*) of phosphorylated and total conventional (pPKC pan), atypical (pPKC $\zeta/\lambda$), novel (pPKC $\delta/\theta$), and PKD/PKC$\mu$ isoforms. Phosphorylated PKC isoforms were normalized by total PKC levels and reported relative to vehicle, n = 3 independent cultures. (F–H) Relationships between vesicular release (pooled data from vesicular release experiments), membrane disruption (pooled from membrane integrity experiments) and ATP release (pooled from 10x tFSS experiments) following mechanical stimulation of CB-OB, *solid line* - regression. For *Figure 6*, data are means ± SEM, *significance compared to vehicle (0 s vehicle for A), [†]significance compared to 300 s vehicle (A) and [#]significance of indicated comparisons (A), by ANOVA or by regression. Source data for *Figure 6* is provided in *Figure 6—source data 1*.

DOI: https://doi.org/10.7554/eLife.37812.027

The following source data and figure supplements are available for figure 6:

**Source data 1.**

DOI: https://doi.org/10.7554/eLife.37812.032

**Figure supplement 1.** PKC regulates membrane resealing and ATP release in C2-OB cell line.

DOI: https://doi.org/10.7554/eLife.37812.028

**Figure supplement 2.** Membrane resealing, vesicular exocytosis and ATP release are Ca[2+]/PLC-dependent processes.

DOI: https://doi.org/10.7554/eLife.37812.033

**Figure supplement 3.** Basal vesicular density and mechanically-stimulated vesicular release in compact bone-derived osteoblasts.

DOI: https://doi.org/10.7554/eLife.37812.029

**Figure supplement 4.** Immunoblot analysis of PKC isoforms.

DOI: https://doi.org/10.7554/eLife.37812.030

**Figure supplement 5.** Effect of drug treatments on [Ca[2+]]i elevations and ATP bioluminescence assay.

DOI: https://doi.org/10.7554/eLife.37812.031

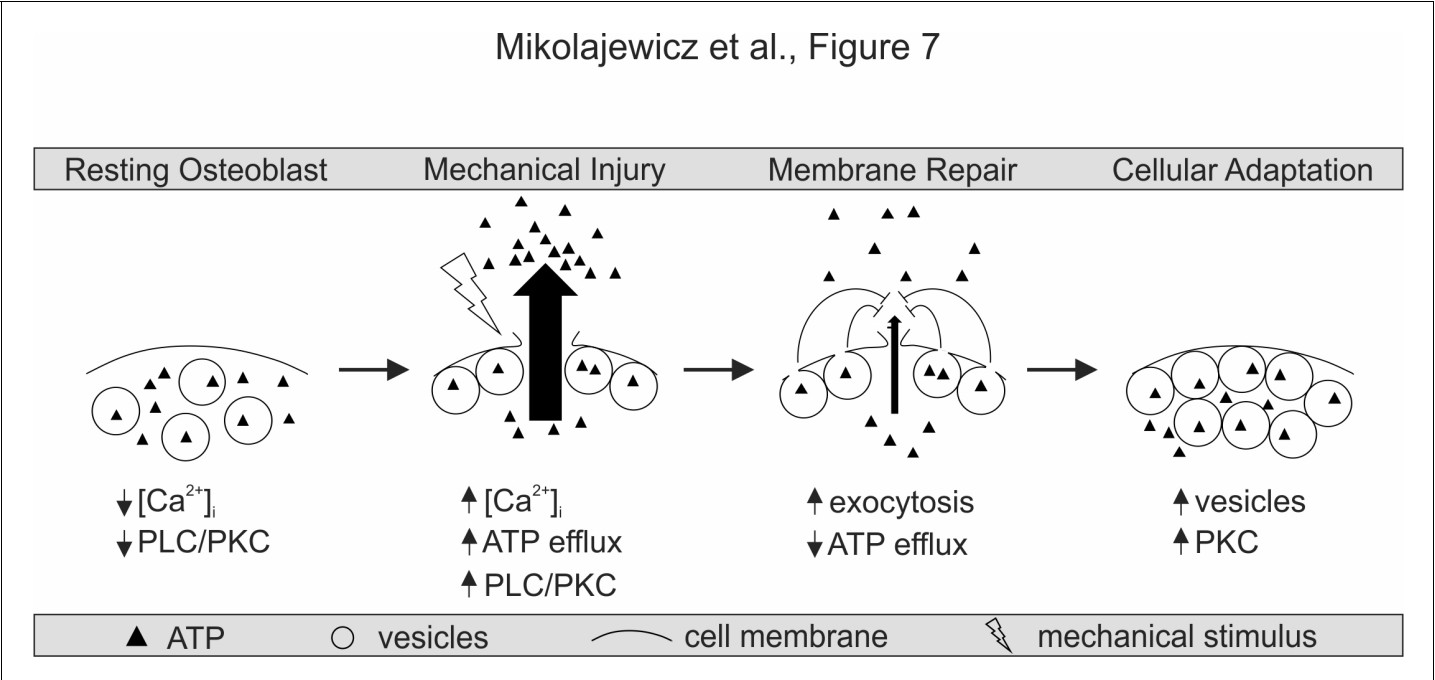

**Figure 7.** Proposed model for mechanically stimulated ATP release in osteoblasts. *Resting Osteoblasts*: Under basal conditions, the cell membrane is intact and intracellular free calcium levels and PLC/PKC signaling are minimal. *Mechanical injury*: Mechanical stimulation results in disruption of the cellular membrane, which leads to the influx of calcium, activation of PLC/PKC signalling, and the efflux of ATP. *Membrane repair*: The disrupted membrane is rapidly repaired through a process involving $Ca^{2+}$/PLC/PKC-dependent vesicular exocytosis, thereby limiting ATP release. *Cellular adaptation*: Elevated PKC levels result in priming of the vesicular pathway, likely regulating cellular resilience and responsiveness to subsequence cycles of mechanical stimulation.

DOI: https://doi.org/10.7554/eLife.37812.024

## Discussion

This study establishes a novel mechanism of mechanotransduction in bone cells, which positions ATP release as the integrated outcome of membrane injury (reflective of the destructive potential of mechanical forces) and membrane repair (countering and adapting to the damage). We report that mechanical stimuli *in vitro* and *in vivo* induced reversible cell membrane disruption. This disruption resulted in ATP release predominantly through the compromised membrane, rather than through vesicular or conductive mechanisms. $Ca^{2+}$/PLC/PKC-dependent vesicular release was necessary for successful membrane repair and prevention of further ATP release. Importantly, prior activation of PKD/PKCμ or vesicular priming improved membrane integrity and repair, thereby minimizing membrane injury and ATP release upon subsequent mechanical stimulation, and serving as a mechanism of cellular mechano-adaptive memory.

### Membrane injury

We have demonstrated that non-lethal cell membrane injury is routine in mechanically stimulated cells. *In vitro*, single cell membrane deformation or fluid shear stress transiently compromised cell membrane integrity in 2–30% of cells, which was repaired within 10–100 s. Consistent with findings observed using *in vitro* models of mechanical stimulation, *in vivo* cyclic compressive tibial loading engendering physiological and supraphysiological strains resulted in transient osteocyte membrane disruption in 5–10% of cells. Interestingly, supraphysiological mechanical strains improved membrane integrity post-stimulation compared to baseline levels. Previously, non-lethal reversible cell wounding was shown to occur in mechanically active tissues, with membrane-impermeable marker uptake reported in 3–20% of cells in skeletal muscle, 3–6% in skin, 25% in cardiac muscle, 2–30% in lung and aorta (*McNeil and Steinhardt, 2003*). Consistent with our findings, membrane disruption was reported in 20–60% of long bone osteocytes after *in vivo* treadmill loading (*Yu et al., 2017*).

The higher prevalence of disruption reported by Yu et al. in long bones was likely a consequence of cumulative tracer uptake over 18 hr (*Yu et al., 2017*), while our study examined membrane disruption over a 20–50 min period. Our data demonstrated that reparable membrane injuries are common and significantly contribute to ATP release from mechanically stimulated osteoblasts and osteocytes.

## ATP release

We observed that osteoblasts release $70 \pm 24$ amol ATP/cell in response to direct membrane deformation and $21 \pm 11$ to $422 \pm 97$ amol ATP/cell in response to tFSS, which is consistent with previously reported estimates ranging from $24 \pm 1$ to $324 \pm 59$ amol ATP/cell (*Genetos et al., 2005*; *Wang et al., 2013*; *Kringelbach et al., 2015*; *Kringelbach et al., 2014*; *Genetos et al., 2007*; *Pines et al., 2003*; *Romanello et al., 2005*; *Manaka et al., 2015*). Osteoblasts have been reported to release ATP through mechanisms related to vesicular exocytosis (*Genetos et al., 2005*; *Romanello et al., 2005*) and L-type VSCC (*Genetos et al., 2005*). In osteocytes, hemichannels (*Kringelbach et al., 2015*; *Seref-Ferlengez et al., 2016*), P2X7 (*Seref-Ferlengez et al., 2016*) and T-type VSCC (*Thompson et al., 2011*) have also been implicated. We visualized ATP-containing quinacrine-positive vesicles and directly demonstrated their release upon mechanical stimulation. Surprisingly, the total vesicular ATP content was $\sim 94 \pm 5$ amol ATP/cell, which, considering that only $7.2 \pm 1.2\%$ of vesicles were released upon mechanical stimulation, is much less ($<10$ amol ATP/cell) than mechanically induced ATP release. Moreover, treatments that potentiated vesicular release resulted in a surprising decrease in ATP release, while treatments interfering with vesicular release enhanced ATP release. In search for an alternative route of ATP release, we found that mechanically stimulated ATP release was proportional to the extent of membrane injury.

## Membrane resealing

Vesicular exocytosis is believed to promote membrane repair via membrane insertion, thereby reducing membrane tension and allowing exposed hydrophobic residues to reseal (*McNeil and Steinhardt, 2003*). We have shown that vesicular release was critical for the repair of mechanically induced membrane injury and that the $Ca^{2+}$/PLC/PKC pathway regulates vesicular exocytosis in osteoblasts. We interrogated the role of vesicular release in the mechanoresponse using NEM, a modulator of vesicular docking machinery (*Lonart and Südhof, 2000*). While NEM is commonly used as an inhibitor of vesicular release (*Genetos et al., 2005*; *Kowal et al., 2015*), it has also been shown to increase vesicular release probability (*Kirmse and Kirischuk, 2006*) through accumulation of readily-releasable vesicles (*Lonart and Südhof, 2000*). Using real-time imaging, we demonstrated that NEM increased the basal vesicular density and potentiated mechanically stimulated vesicular exocytosis in osteoblasts. In the presence of NEM, mechanically induced membrane injury and ATP release were significantly reduced. Vesicular exocytosis has been implicated as a mechanism of ATP release from osteoblasts and osteocytes (*Genetos et al., 2005*; *Kringelbach et al., 2015*; *Romanello et al., 2005*) based on the assumed inhibition of vesicular exocytosis by NEM or vesicular trafficking by brefeldin A, bafilomycin or monensin. However, our data demonstrated that the role of vesicles cannot be inferred without a direct validation of the effect of NEM on vesicular release. In our study, $[Ca^{2+}]_e$ depletion abolished mechanically stimulated vesicular release, interfered with membrane repair and increased ATP release. Calcium is one of the first mechanically induced signals that initiates downstream responses (*Robling and Turner, 2009*), including exocytosis-mediated membrane repair (*Cooper and McNeil, 2015*). It was previously shown that during *in vivo* loading of mouse bone, the number of osteocytes exhibiting $[Ca^{2+}]_i$ elevations increased proportionally to the magnitude of applied strain (*Lewis et al., 2017*). We found that PLC and PKCμ, known downstream targets of $Ca^{2+}$ signaling (*Mochly-Rosen et al., 2012*), regulated basal vesicle abundance and membrane resealing. Our study suggests that the osteoporotic phenotype reported in PKCμ-deficient bones (*Li et al., 2017*) may be associated with defective mechanoresponsiveness. Consistent with mechanisms of facilitated membrane resealing (*Togo et al., 1999*; *Cooper and McNeil, 2015*), we propose that mechanical stimulation results in membrane disruption, which leads to ATP efflux and $Ca^{2+}$ influx, triggering vesicular exocytosis and membrane repair that then limits ATP release (*Figure 7*).

## Mechano-adaptive memory

Our findings suggest that osteoblast mechano-adaptive status may be influenced by PKCµ/vesicular signaling. Activation of PKC prior to mechanical stimulation or priming the vesicular pool for release with NEM resulted in drastic decreases in membrane injury and ATP release in response to subsequent mechanical stimulation. Prior work has demonstrated that mechanical loading induces the production and release of LAMP-1-positive vesicles from osteocytes (LAMP-1 colocalizes with the quinacrine tracer-dye used in this study (*Cao et al., 2014*), which was critical for mechano-adaptive bone formation (*Morrell et al., 2018*). This mechano-adaptive effect may underlie cellular accommodation in the bone response to mechanical loading previously reported (*Schriefer et al., 2005*), and explain the reduced responsiveness of osteocytes exposed to higher frequency *in vivo* mechanical loading, compared to lower frequency loading of the same magnitude (*Lewis et al., 2017*). Our findings suggest that during repeated mechanical stimulation, the priming of PKCµ/vesicular pathway by earlier stimuli will determine osteoblast resilience and responsiveness to subsequent cycles of mechanical stimulation, representing a mechanism for cellular accommodation.

## Concluding remarks

In this study, we established the regulatory mechanisms involved in controlling the amount of ATP released following reversible cellular injury. We have demonstrated that membrane injury and repair are common under physiological stresses *in vitro* and *in vivo*, and determined the critical role for PKC-regulated vesicular exocytosis in bone cell membrane repair. We suggest that rather than delivering ATP to extracellular space, exocytosis of ATP-containing vesicles limits the much larger efflux of intracellular ATP through damaged membranes. We propose a model of biological adaptation to mechanical forces, which directly links mechanosensation through reversible membrane injury and ATP release to the development of adaptive resilience against the destructive potential of mechanical forces. PKC-mediated vesicular release provides a target for potential therapeutic interventions to modulate mechano-responsiveness and mechano-resilience in humans habitually experiencing altered mechanical environment, such as astronauts or paralysis patients.

# Materials and methods

All procedures were approved by McGill's University's Animal Care Committee (protocols # 2012–7127 and 2016–7821) and complied with the ethical guidelines of the Canadian Council on Animal Care.

**Key resources table**

| Reagent type (species) or resource | Designation | Source or reference | Identifiers | Additional information |
|---|---|---|---|---|
| Cell line (*M. musculus*) | BMP2-transfected C2C12 myoblast cells (C2C12-BMP2; C2-OB) | Dr. M. Murshed, McGill University | | osteoblast cell line |
| Strain, strain background (*M.musculus*, female) | 10 week-old C57Bl/6J | Jackson Laboratory | Stock #: 000664; RRID: IMSR_JAX:000644 | primary osteoblast source, *in vivo* studies |
| Software, algorithm | Calcium Analyser | DOI: 10.3389/ fphys.2016.00525 | | MATLAB algorithm for calcium analysis |
| Chemical compound, drug | MANT-ATP | AnaSpec | Cat. AS-64610 | fluorescent ATP analog, 5 µM (overnight, 37°C) |
| Chemical compound, drug | GsMTx-4; GsM | Alomone Labs | Cat. STG-100 | Piezo1 inhibitor, 1 µM (10 min, room temperature) |
| Chemical compound, drug | U73122; U73 | Calbiochem | Cat. 662035 | PLC inhibitor, 10 µM (10 min, room temperature) |

*Continued on next page*

*Continued*

| Reagent type (species) or resource | Designation | Source or reference | Identifiers | Additional information |
|---|---|---|---|---|
| Antibody | GAPDH (D16H11, Rabbit antibody) | Cell Signaling Technology | Cat. 5174; RRID: AB_10622025 | WB, 1:1000 dilution, 5% BSA/TBST (1 hr, room temperature) |
| Antibody | β-tubulin (9F3, Rabbit antibody) | Cell Signaling Technology | Cat. 2128; RRID: AB_823664 | WB, 1:1000 dilution, 5% BSA/TBST (1 hr, room temperature) |
| Antibody | p-PKC (pan; βII Ser660, Rabbit antibody) | Cell Signaling Technology | Cat. 9371; RRID: AB_330848 | WB, 1:1000 dilution, 5% BSA/TBST (over night, 4°C) |
| Antibody | PKCα (Rabbit antibody) | Cell Signaling Technology | Cat. 2056; RRID: AB_2284227 | WB, 1:1000 dilution, 5% BSA/TBST (over night, 4°C) |
| Antibody | p-PKCδ/θ (Ser643/676, Rabbit antibody) | Cell Signaling Technology | Cat. 9376; RRID: AB_330848 | WB, 1:1000 dilution, 5% BSA/TBST (over night, 4°C) |
| Antibody | PKCδ (D10E2, Rabbit anitbody) | Cell Signaling Technology | Cat. 9616; RRID: AB_10949973 | WB, 1:1000 dilution, 5% BSA/TBST (over night, 4°C) |
| Antibody | p-PKD/PKCμ (Ser744/748, Rabbit antibody) | Cell Signaling Technology | Cat. 2054; RRID: AB_330848 | WB, 1:1000 dilution, 5% BSA/TBST (over night, 4°C) |
| Antibody | PKD/PKCμ (D4J1N, Rabbit antibody) | Cell Signaling Technology | Cat. 90039 | WB, 1:1000 dilution, 5% BSA/TBST (over night, 4°C) |
| Antibody | p-PKCζ/λ (Thr410/403, Rabbit antibody) | Cell Signaling Technology | Cat. 9378; RRID: AB_330848 | WB, 1:1000 dilution, 5% BSA/TBST (over night, 4°C) |
| Antibody | PKCζ (C24E6, Rabbit antibody) | Cell Signaling Technology | Cat. 9368; RRID:AB_10693777 | WB, 1:1000 dilution, 5% BSA/TBST (over night, 4°C) |
| Antibody | anti-rabbit IgG, HRP-linked secondary antibody) | Cell Signaling Technology | Cat. 7074; RRID:AB_2099233 | WB, 1:1000 dilution, 5% BSA/TBST (1 hr, room temperature) |
| Chemical compound, drug | Minimum Essential Medium (MEM) α; αMEM | Gibco | Cat. 12,000–022 | cell culture reagent |
| Chemical compound, drug | Trypan Blue Solution, 0.4%; TB | Gibco | Cat. 15250061 | *in vitro* injury assay dye |
| Commercial assay or kit | Alamar Blue Cell Viability Reagent | Invitrogen | Cat. DAL1025 | cell viability assay |
| Chemical compound, drug | DAPI | Invitrogen | Cat. D1306; RRID:AB_2629482 | IF, nucleur stain, 1:5000 dilution, $H_2O$ (5 min, room temperature) |
| Chemical compound, drug | Fura2 AM | Invitrogen | Cat. F1221 | ratiometric calcium-binding dye (30 min, room temperature) |
| Chemical compound, drug | D-Luciferin potassium salt | Invitrogen | Cat. L2916 | luciferase substrate, ATP bioluminescence assay |
| Chemical compound, drug | Lucifer Yellow CH, Lithium Salt; LY | Invitrogen | Cat. L453 | GAP junction permeable dye, 10 μM (2 min, 37°C) |
| Chemical compound, drug | Rhodamine B-conjugated 10 000 MW Dextran; R-dextran | Invitrogen | Cat. D1824 | *in vitro* injury assay dye |

*Continued on next page*

Continued

| Reagent type (species) or resource | Designation | Source or reference | Identifiers | Additional information |
|---|---|---|---|---|
| Chemical compound, drug | lysine-fixable Texas Red-conjugated 10 000 MW Dextra; LFTR-Dex | Invitrogen | Cat. D1863 | *in vivo* injury assay dye |
| Antibody | Sclerostin (Goat antibody) | R and D Systems | Cat. AF1589; RRID:AB_2195345 | IF, 1:1000 dilution, 5% BSA/TBST (over night, 4°C) |
| Chemical compound, drug | Collagenase P from *Clostridium histolyticum* | Roche | Cat. 11213857001 | enzymatic digest of bone fragments |
| Commercial assay or kit | Cytotoxicity Detection Kit$^{PLUS}$; LDH | Roche | Cat. 04 744 926 011 | LDH leakage assay |
| Antibody | Donkey anti-goat IgG-FITC | Santa Cruz | Cat. Sc-2024; RRID:AB_631727 | IF, 1:1000 dilution, 5% BSA/TBST (1 hr, room temperature) |
| Chemical compound, drug | Adenosine 5'-triphosphate magnesium salt; ATP | Sigma-Aldrich | Cat. A9187 | |
| Chemical compound, drug | Bisindolylmaleimide II; Bis | Sigma-Aldrich | Cat. B3056 | PKC inhibitor, 1 µM (10 min, room temperature) |
| Chemical compound, drug | Carbenoxolone disodium salt; Cbx | Sigma-Aldrich | Cat. C4790 | Hemichannel blocker, 10 µM (10 min, room temperature) |
| Chemical compound, drug | Fast Red Violet LB Salt | Sigma-Aldrich | Cat. F3381 | alkaline phosphatase assay reagent |
| Chemical compound, drug | Gadolinium (III) chloride; Gd$^{3+}$ | Sigma-Aldrich | Cat. 439770 | Mechanically-sensitive channel blocker, 10 µM (10 min, room temperature) |
| Chemical compound, drug | GSK2193874; GSK | Sigma-Aldrich | Cat. SML0942 | TRPV4 inhibitor, 100 nM (10 min, room temperature) |
| Chemical compound, drug | HC-067047; HC | Sigma-Aldrich | Cat. SML0143 | TRPV4 inhibitor, 100 nM (10 min, room temperature) |
| Chemical compound, drug | L-ascorbic acid 2-phosphate sesquimagnesium salt hydrate | Sigma-Aldrich | Cat. A8960 | osteoblast differentiation reagent, 50 µg/mL |
| Chemical compound, drug | Luciferase from *Photinus pyralis* | Sigma-Aldrich | Cat. L9420 | ATP bioluminescence assay reagent |
| Chemical compound, drug | ML218; ML | Sigma-Aldrich | Cat. SML0385 | T-type VSCC inhibitor, 10 µM (10 min, room temperature) |
| Chemical compound, drug | Naphthol AS-MX phosphate disodium salt | Sigma-Aldrich | Cat. N5000 | alkaline phosphatase assay reagent |
| Chemical compound, drug | N-ethylmaleimide; NEM | Sigma-Aldrich | Cat. E3876 | Vesicular activator, 1 mM (10 min, room temperature) |
| Chemical compound, drug | Nifedipine; Nif | Sigma-Aldrich | Cat. N7634 | L-type VSCC inhibitor, 10 µM (10 min, room temperature) |
| Chemical compound, drug | 1-octanol; Oct | Sigma-Aldrich | Cat. 297877 | Connexin blocker, 1 mM (10 min, room temperature) |

*Continued on next page*

*Continued*

| Reagent type (species) or resource | Designation | Source or reference | Identifiers | Additional information |
|---|---|---|---|---|
| Chemical compound, drug | phorbol 12-myristate 13-acetate; PMA | Sigma-Aldrich | Cat. P8139 | PKC activator, 100 nM (10 min, room temperature) |
| Chemical compound, drug | Quinacrine dihydrochloride | Sigma-Aldrich | Cat. Q3251 | Vesicle dye, 10 μM (15 min, room temperature) |
| Chemical compound, drug | Flufenamic Acid; FFA | Tocris Bioscience | Cat. 4522 | Hemichannel blocker, 10 μM (10 min, room temperature) |
| Chemical compound, drug | Ionomycin calcium salt; Iono | Tocris Bioscience | Cat. 1704 | calcium ionophore, 100 μM |
| Chemical compound, drug | PPADS tetrasodium salt; PPADS | Tocris Bioscience | Cat. 0625 | P2R inhibitor, 100 μM (10 min, room temperature) |
| Chemical compound, drug | Suramin hexasodium salt; Sur | Tocris Bioscience | Cat. 1472 | P2R inhibitor, 100 μM (10 min, room temperature) |
| Chemical compound, drug | A 740003; A7 | Tocris Bioscience | Cat. 3701 | P2X7 inhibitor, 1 μM (10 min, room temperature) |
| Chemical compound, drug | Dulbecco's Modified Eagle Medium; DMEM | Wisent Bio Products | Cat. 319–020 CL | cell culture reagent |
| Chemical compound, drug | Fetal bovine Serum; FBS | Wisent Bio Products | Cat. 080152 | cell culture reagent |
| Chemical compound, drug | Penicillin Streptomycin | Wisent Bio Products | Cat. 450–201-EL | antibiotic, 1% |
| Chemical compound, drug | Sodium Pyruvate | Wisent Bio Products | Cat. 600–110 UL | cell culture reagent |
| Chemical compound, drug | Collagenase Type II | Worthington Biochemical Corporation | Cat. LS004176 | enzymatic digest of bone fragments |

## Solutions

Phosphate-buffered saline (PBS; 140 mM NaCl, 3 mM KCl, 10 mM $Na_2HPO_4$, 2 mM $KH_2PO_4$, pH 7.4), sterilized by autoclave. Physiological solution (PS; 130 mM NaCl; 5 mM KCl; 1 mM $MgCl_2$; 1 mM $CaCl_2$; 10 mM glucose; 20 mM HEPES, pH 7.6), sterilized by 0.22 μM filtration. In $[Ca^{2+}]$-free physiological solution, $CaCl_2$ was excluded and 10 mM EGTA was added to chelate any residual calcium, sterilized by 0.22 μm filtration. Bioluminescence reaction buffer (0.1 M DTT, 25 mM tricine, 5 mM $MgSO_4$, 0.1 mM EDTA, 0.1 mM $NaN_3$, pH 7.8), sterilized by 0.22 μm filtration. RIPA lysis buffer (50 mM Tris, pH 7.4, 150 mM NaCl, 1% Nonidet P-40, 1 mM EDTA, 1 mg/mL aprotinin, 2 mg/mL leupeptin, 0.1 mM phenylmethylsulfonyl fluoride, 20 mM NaF, 0.5 mM $Na_3VO_4$). TBST buffer (10 mM Tris-HCL, pH 7.5, 150 mM NaCl, 1% Tween 20).

## Pharmacological interventions

The pharmacological interventions (final concentration, name (*abbreviation*): molecular target) used in this study were: 10 μM Gadolinium (**Gd³⁺**): maxi-anion channel, 10 μM Carbenoxolone (**Cbx**): connexins/pannexins, 10 μM Flufenamic Acid (**FFA**): connexins, 1 mM 1-Octanol (**Oct**): connexins, 100 nM GSK 2193874 (**GSK**): TRPV4, 100 nM HC 067047 (**HC**): TRPV4, 10 μM Nifedipine (**Nif**): L-type VSCC, 10 μM ML218 (**ML**): T-type VSCC, 100 μM Suramin (**Sur**): P2 receptors (P2R), 100 μM **PPADS**: P2 receptors, 1 μM A740003 (**A7**): P2X7 receptor, 10 μM U73122 (**U73**): PLC, 100 nM Phorbol 12-myristate 13-acetate (**PMA**): PKC, 1 μM Bisindolylmaleimide II (**Bis**): PKC, 1 mM N-Ethylmaleimide (**NEM**): NSF, 1 μM GsMTx-4 (**GsM**): Piezo1. Unless stated otherwise, cells were treated with drugs for 10 mins prior to experiments. We validated the effects of hemichannel blockers (*Figure 4—figure supplement 1*) and evaluated the effects of the inhibitors on ATP- or citrate-induced $[Ca^{2+}]_i$

responses in Fura2-loaded C2-OB cells and on luciferin/luciferase bioluminescence (*Figure 6—figure supplement 5*).

## Cell cultures

Three osteoblast models were used in this study. All key experiments required to support our proposed model were performed using primary CB-OB cells; however, additional data from another primary source (bone-marrow-derived; BM-OB) and cell-line (C2-OB) were reported to emphasize the validity of our findings and demonstrate how different osteoblast models perform in this study.

### Bone-marrow-derived osteoblasts (BM-OB)

To obtain bone-marrow-derived osteoblasts, 4-6 week C57Bl/6J mice were sacrificed and femur and tibia were collected. Bone pieces were sterilized in 70% EtOH and washed (3x, PBS). Bones were cut in half and placed into longitudinally-cut 1 mL pipette tips, with the exposed marrow end of the bone facing down. Pipette tips were centrifuged in Eppendorf tubes at 12 000 rpm (3x, 30 s) to isolate bone marrow cells. Bone marrow cells were plated at a density of $5\times10^4$ cells per cm$^2$ in osteoblastic differentiation medium (αMEM supplemented with 10% FBS, 1% sodium pyruvate, 1% penicillin streptomycin and 50 µg/mL ascorbic acid) and media was refreshed every 2-3 days. Osteoblast phenotype was observed within 7-11 days.

### Compact bone-derived osteoblasts (CB-OB)

Bone-marrow-depleted bones, obtained after bone marrow cell isolation, were used as source of CB-OB. Epiphyseal ends of the bone were cut off and the remaining diaphysis were cut into small fragments (<1 mm$^2$ pieces). Bone fragments underwent a three-step enzymatic digest: (i) 10 mL αMEM (serum-free) + 125 µL 0.25% trypsin + 5 µL collagenase P (37°C, 15 min while rotating). (ii) Dispose supernatant and add 10 mL αMEM (serum-free) + 125 µL 0.25% trypsin + 10 µL 100 mg/mL collagenase P (37°C, 30 min while rotating). (iii) Dispose supernatant and add 10 mL αMEM (serum-free) + 125 µL 0.25% trypsin + 100 µL 100mg/mL collagenase II (37°C, 1 hr while rotating, shake vigorously every 10 min). Supernatant from final digest was disposed, bone fragments were rinsed with PBS and plated in 6-well plate in αMEM (10% FBS, 1% sodium pyruvate, 1% penicillin streptomycin and 50 µg/mL ampicillin) for 3-5 days to allow cell outgrowth from fragments. Cells were detached with trypsin, filtered through 40 µm membrane and plated at a density of $1\times10^4$ per cm$^2$ in osteoblastic differentiation medium (αMEM supplemented with 10% FBS, 1% sodium pyruvate, 1% penicillin streptomycin and 50 µg/mL ascorbic acid) 2-3 days prior to experiments. Primary osteoblast cultures were sub-cultured for up to 8 passages (1:10 passages).

### BMP-2 tranfected C2C12 cells (C2-OB)

The C2C12 cell line (ATCC CRL-1772) stably transfected with BMP-2 (courtesy of Dr. M. Murshed, McGill University) was maintained in DMEM (supplemented with 1% penicillin streptomycin, 1% sodium pyruvate and 10% FBS) at 5% CO$_2$, 37°C humidified atmosphere. Absence of mycoplasma contamination was verified in cryo-preserved stocks of C2-OB cells using the Venor GeM Mycoplasma PCR-based detection kit (Sigma MP0025). To further authenticate the osteoblast phenotype in the C2-OB cell line, osteogenic differentiation was induced with ascorbic acid (50 µg/mL) and changes in osteogenic gene expression after 7 days of culture were assessed by qRT-PCR. Significant increases in expression of osteoblast marker genes Runx2 (+37%, p=0.02), *Alpl* (+57%, p=0.03) and *Col1a1* (+172%, p<0.001) were observed, while *Bmp2* expression was stable overtime, thereby confirming the osteoblastic phenotype in this cell-line. Cells between passages 6 and 10 were used in all experiments. 2-3 days prior to experiments, cells were plated at a density of $3\times10^4$ cells per cm$^2$.

## Osteoblast phenotype

Cells were fixed with formalin (pH 7.4, 8 min) and rinsed with PBS (3x). Alkaline phosphatase (ALP) staining solution was prepared by combining 4.5 mg fast red violet salt dissolved in 3.75 mL H$_2$O and 3.75 mL 0.2 M Tris-HCl, pH 8.3 with 0.75 mg Naphthol AS-MX phosphate disodium salt dissolved in 30 µL N,N-dimethylformamide. Sufficient staining solution was added to each well to ensure entire surface was covered (approx. 500 µL in 35 mm dish) and incubated at room

temperature for 15 min, or until red precipitate formed. Cells were washed with $H_2O$ and observed under bright-field microscopy. Alkaline phosphate-positive cells were stained pinkish-red, while yellow hue was observed in alkaline phosphatase-negative cells.

## Mechanical stimulation

### Local membrane deformation of a single cell

Single osteoblastic cells were stimulated by local membrane deformation with a glass micropipette using a FemtoJet microinjector NI2 (Eppendorf Inc.). The micropipette was positioned approximately 10 μm from the cell membrane at a 45° angle from the horizontal plane and mechanically stimulated at a speed of 250 μm/s with a contact duration of 60 ms. Glass micropipettes were custom-made from glass capillary tubes (1.5 mm outer diameter, King Precision Glass Inc.) using a Sutter Pipette Puller (Flaming/Brown, model P-87).

### Bulk stimulation by turbulent fluid shear (tFSS)

50% of the culture medium volume was manually withdrawn and replaced at an approximate speed of 2 resuspensions per second for an indicated number of $n$ times (e.g., 10x tFSS corresponded to 50% vol displacement performed 10 times).

## Intracellular calcium recording and analysis

Cells plated on glass-bottom plates (MatTek Corporation) were loaded with a ratiometric fluorescent calcium dye Fura2-AM (r.t., 30 min), washed with PS and allowed to acclimatize for 10 min to reduce the effects of mechanical agitation. $[Ca^{2+}]_i$ was recorded at a sampling rate of 2 images per second (2 Hz) using a Nikon T2000 fluorescent inverted microscope with 40x objective (*Tiedemann et al., 2009*). Recordings consisted of ~10 s baseline $[Ca^{2+}]_i$ and additional 110–170 s following application of mechanical stimulation or treatment. The excitation wavelength was alternated between 340 and 380 nm using an ultra-high-speed wavelength switching illumination system (Lambda DG-4, Quorum Technologies). Regions of interest (ROI) were manually defined and the ratio of the fluorescent emission at 510 nm, following 340 and 380 nm excitation (f340/f380), was calculated and exported using the imaging software (Volocity, Improvision). All data were imported into an excel spreadsheet for characterization using a MATLAB algorithm previously described (*Mackay et al., 2016*). The following parameters were obtained for statistical analysis: amplitude (amp; magnitude of response), activation time ($t_{10\%-90\%}$; time between 10% and 90% of maximum response), and decay constant ($\tau_{decay}$) of exponential decay region of deactivation phase of transient response. For micropipette stimulation experiments, secondary responsiveness was calculated as the percentage of neighboring cells exhibiting $[Ca^{2+}]_i$ elevations.

## Vesicle labeling, imaging, and analysis

### Localization

To visualize the intracellular distribution and colocalization of quinacrine and MANT-ATP, cells were loaded with MANT-ATP (5 μM, overnight, 37°C) followed by quinacrine (10 μM, 15 min, room temperature). Cells were washed with PS and imaged using a Zeiss LSM 780 laser scanning confocal microscope. An argon and diode laser were used as light sources (ex. 405 nm and 458 nm) and the detection wavelength ranges were set to 410–470 nm and 490–570 nm to ensure separation of overlapping emission spectra, for MANT-ATP and quinacrine, respectively. Images were acquired using a Zeiss Plan-Apochromat 63x/1.40 Oil immersion objective lens. Subsequent analysis was carried out using ImageJ (National Institutes of Health, USA). Vesicular density was determined using ImageJ by counting number of intracellular quinacrine-positive puncta and normalizing count to cell surface area.

### Kinetics

To study vesicular release kinetics, time-lapse recordings of quinacrine-loaded cells were acquired with a Nikon T2000 fluorescent inverted microscope after a 10-min period of post-stain acclimatization to minimize the effects of mechanical agitation. Quinacrine-rich vesicles were imaged using a 40x objective at a sampling frequency of 2 Hz for 2 min. To minimize the phototoxic effects of quinacrine, the excitation intensity was set to 33% of peak intensity and the shutter was shut in between

100 ms exposures. Image sequences were exported using the imaging software Volocity (Improvision) for subsequent analysis in ImageJ. To quantify the kinetics of vesicular release, sudden losses of fluorescence were monitored. Vesicles that gradually came in and out of focus did not qualify as release events. SparkSpotter (Babraham Bioinformatics, UK), an ImageJ plugin, was used to identify release events in temporally reversed and contrast-enhanced image stacks, so that the loss of fluorescence would instead be detected as a sudden increase by the spark identifying algorithm. Acquired data was visually evaluated and represented as either release events per second or cumulative vesicular release for indicated time frame, normalized to unit area.

## ATP measurements

### Fluorescence recordings of $[ATP]_e$

10 nM D-luciferin- and 2 µg/mL fire-fly luciferase-containing physiological solution (PS) was added to C2-OB and extracellular luciferin fluorescence was recorded (380 ex/510 em). Using an ATP standard calibration curve (*Figure 2—figure supplement 1*), pericellular (<15 µm from stimulated cell) ATP concentrations [ATP] were obtained. The amount of ATP released by individual cells was estimated by fitting the diffusion profile obtained from the fluorescence measurements with the diffusion equation for release from an instantaneous plane source (*Crank, 1975*).

$$[ATP](x,t) = \frac{M}{2(\pi Dt)^{\frac{1}{2}}} \exp\left(\frac{-x^2}{4Dt}\right)$$

where $[ATP](x,t)$ is ATP concentration at distance $x$ from the source, at time $t$ from the moment of release. $M$ is the amount of ATP released at $x = 0$, $t = 0$. $D$ is ATP diffusion coefficient, assumed to be $3.3 \times 10^{-6}$ cm$^2$/s at 22°C (*Newman, 2001*).

### Bioluminescent recordings of $[ATP]_e$

Bioluminescence assay was used to measure ATP content in supernatant samples. Luciferin-luciferase reaction solution contained 0.45 µM D-luciferin and 40 µg/mL firefly-luciferase in bioluminsecence reaction buffer. 10 µL supernatants were sampled after tFSS stimulation, added into a glass cuvette containing 100 µL of luciferin-luciferase reaction solution, and measured using FB 12 single tube luminometer. Total cellular ATP content was measured in RIPA-lysed samples. [ATP] was estimated using calibration curves (*Figure 6—figure supplement 5*).

## Membrane integrity assays

### Dye leakage

Fura2-loaded osteoblastic cells were mechanically stimulated with a glass micropipette and 510 nm emission intensity was monitored after 340 nm excitation (*Togo et al., 1999*). Unless stated otherwise, Fura2 leakage refers to the percentage of cells that were severely injured (sIn; complete loss of intracellular Fura2 fluorescence) following micropipette stimulation.

### Dye uptake

0.08% TB or 10 µM R-dextran were added before or 5 min after mechanical stimulation, washed with PS, and the proportion of dye-positive cells were assessed by microscopy.

### Cytosolic leakage

Extracellular lactate dehydrogenase (LDH) was measured using Cytotoxicity Detection Kit$^{PLUS}$. Total cellular LDH content was measured in RIPA-lysed samples.

### Cell viability

1 hr after tFSS, AlamarBlue assay was performed according to manufacturer's protocol. Cell viability was reported as percentage of total cells.

## *In vivo* bone loading

10-week-old female C57Bl/6J mice were injected intraperitoneally with LFTR-Dex either 30 min before or 20 min after loading. *In vivo* cyclic compressive loading at 600 µε or 1200 µε was applied

for 5 min (*Willie et al., 2013*). Mice were randomized into two groups, anesthetized, and *in vivo* cyclic compressive loading was applied to the left tibia (216 cycles at 4 Hz the mean mouse locomotor stride frequency [*Clarke and Still, 1999*]), delivering a maximum force of −5.5 N and −11 N, which engenders 600 με or 1200 με, respectively at the periosteal surface of the tibia mid-diaphysis in these mice determined by prior *in vivo* strain gauging studies (*Birkhold et al., 2014*). Previous reports have shown strain levels of 200–600 με are engendered on the medial tibia during walking in the mouse (*De Souza et al., 2005*; *Sugiyama et al., 2012*). Strain levels of 1200 με are considered supraphysiological resulting in a robust bone formation response after 5 days of mouse tibial loading (*Willie et al., 2013*; *Brodt and Silva, 2010*) and altered gene expression after only a single bout of loading (*Zaman et al., 2010*; *Kelly et al., 2016*). Mice ambulated freely after controlled loading, and were euthanized 50 min post-dye-injection, weighed (*Figure 5—figure supplement 1A*) and tibiae and calvariae were dissected and fixed in formalin. Tibia lengths were measured (*Figure 5—figure supplement 1B*) and mid-shaft tibiae were cut into fragments, which were immunofluorescently labeled for sclerostin, counterstained with DAPI, and visualized with Nikon T2000 fluorescence inverted microscope. Right non-loaded tibiae were used as an internal control. Female mice were used instead of males to minimize in-cage aggression and related variations in background mechanical-loading. Experimenter was blinded during image analysis.

## Immunoblotting

Cell lysates were extracted in RIPA lysis buffer and centrifuged (12 000 rpm, 10 min, 4°C). Supernatants were collected and protein concentration were determined using Quant-iT protein assay kit (Invitrogen). 70 μg cell lysates were separated on a 10% SDS-PAGE and transferred to a nitrocellulose membrane (0.45 μm, 162–0115, Bio-Rat) using a 10 mM sodium borate buffer. The membranes were blocked in 5% BSA in TBST buffer (1 hr, room temperature) followed by overnight incubation at 4°C with primary antibodies (1:1000 dilution, 5% BSA in TBST) for phosphorylated PKC isoforms: p-PKC (pan; βII Ser660), p-PKCδ/θ (Ser643/676), p-PKD/PKCμ (Ser744/748) and p-PKCζ/λ (Thr410/403), or for total PKC isoforms: PKCα, PKCδ, PKD/PKCμ and PKCζ. Blots were washed and incubated with horseradish peroxidase (HRP)-conjugated secondary antibodies for 1 hr at room temperature (1:1000 dilution, 5% BSA in TBST) and visualized with a chemiluminescence system.

## GAP junction functional assay

Cells were scraped in the presence of GAP-junction permeable dye Lucifer yellow (LY, 10 μM), incubated for 2 min (37°C), rinsed and fixed with formalin (pH 7.4, 8 min). Dye transfer was visualized using wide-field epifluorescence microscopy and cell-cell coupling was visually determined by bright-field microscopy. Cells that were not initially scraped but were physically coupled and LY-positive after 2 min of staining were reported as a percentage of total cells that were physically coupled to scraped neighbors.

## Statistical analysis

Data are representative results or means ± standard errors (S.E.M). Curve fittings and $[Ca^{2+}]_i$ transient characterization was done in MATLAB (MathWorks). Sample sizes for *in vitro* experiments indicate the total number of biological replicates (specified in figure legends as number of stimulated cells or number of independent cultures, which were isolated from different mice for primary OB, or represent different plating dates for C2-OB) pooled across a minimum of three independent experiments. For *in vivo* experiments, sample sizes indicate number of animals. Statistical significance was assessed by ANOVA followed by Bonferroni post-hoc test, significance levels are reported as single symbol ($*p<0.05$), double symbol ($**p<0.01$) or triple symbol ($***p<0.001$).

## Acknowledgements

Confocal imaging was performed at the RI-MUHC Molecular Imaging Platform (McGill University). Special thanks to Dr. M. Murshed (McGill University) for C2-OB cells and M. Rummler and C. Julien (Shriners Hospital for Children-Canada) for help with *in vivo* loading.

## Additional information

### Funding

| Funder | Grant reference number | Author |
|---|---|---|
| Canadian Institutes of Health Research | MOP-77643 | Svetlana V Komarova |
| Natural Sciences and Engineering Research Council of Canada | RGPIN-288253 | Svetlana V Komarova |

The funders had no role in study design, data collection and interpretation, or the decision to submit the work for publication.

### Author contributions

Nicholas Mikolajewicz, Conceptualization, Data curation, Formal analysis, Validation, Investigation, Visualization, Methodology, Writing—original draft, Writing—review and editing; Elizabeth A Zimmermann, Formal analysis, Methodology, Writing—review and editing; Bettina M Willie, Data curation, Formal analysis, Investigation, Methodology, Writing—review and editing; Svetlana V Komarova, Conceptualization, Data curation, Supervision, Funding acquisition, Writing—original draft, Project administration, Writing—review and editing

### Author ORCIDs

Nicholas Mikolajewicz http://orcid.org/0000-0002-7525-0384
Bettina M Willie http://orcid.org/0000-0003-2907-3580
Svetlana V Komarova http://orcid.org/0000-0003-3570-3147

### Ethics

Animal experimentation: All procedures were approved by McGill's University's Animal Care Committee and complied with the ethical guidelines of the Canadian Council on Animal Care. (protocols # 2012-7127 and 2016-7821)

### Decision letter and Author response

Decision letter https://doi.org/10.7554/eLife.37812.034
Author response https://doi.org/10.7554/eLife.37812.035

## Additional files

### Supplementary files

• Transparent reporting form
DOI: https://doi.org/10.7554/eLife.37812.025

### Data availability

All data generated or analysed during this study are included in the manuscript and supporting files. Source data files have been provided for Figures 1-6. Representative video recordings and confocal z-stacks are included as rich media files.

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
