## [Decision Letter]

Thank you for submitting your article "Mechanically-stimulated ATP release from murine bone cells is regulated by a balance of injury and repair" for consideration by *eLife*. Your article has been reviewed by three peer reviewers, and the evaluation has been overseen by Miguel A. Valverde as Guest Reviewing Editor and Vivek Malhotra as the Senior Editor. The following individuals involved in review of your submission have agreed to reveal their identity: Margarita Calvo (Reviewer #2); Paul McNeil (Reviewer #3).

The reviewers have discussed the reviews with one another and the Reviewing Editor has drafted this decision to help you prepare a revised submission.

Overall, this is an interesting manuscript that seeks to explain the mechano-adaptive mechanism of bone cells to repair an injured plasma membrane. The data support a model whereby mechanical stress triggers a response involving ATP release through the disrupted membrane and calcium/PKC dependent vesicular exocytosis to repair the membrane. This is a novel mechanism based on a combination of imaging and functional assays to evaluate the response of both in vitro and in vivo bone cells.

Summary:

Mechanical signals play a critical role in the cellular/tissue regulation of the osteorticular system. Despite the importance of the mechanical loading in osteoarticular system health and disease, the mechanisms that participate in the maintenance of the integrity of the individual cells exposed to cyclical mechanical loading are not fully understood. Recently, different mechanoosmosensitive ion channels have been implicated in the homeostatic response to mechanical loading. The manuscript by Mikolajewicz and colleagues use different cultured bone cells as well as in vivo assays to address the how non-lethal cell membrane injury triggers cell membrane repair. They show that mechanical stress, generates ATP release, intracellular calcium signals and Ca^2+^/PKC dependent membrane exocytosis.

The authors build a model in which the mechanical stimulation results in membrane disruption, which leads to ATP efflux and Ca^2+^ influx, triggering vesicular exocytosis and membrane repair that then limits ATP release. Although it seems a convincing model we would like to see further work done to evaluate the contribution of mechanosensitive ion channels in the repairing mechanism.

Overall, this manuscript is well written (but with overcrowded figures), interesting, timely and will add an interesting angle to the mechanisms of cellular response to mechanical loading.

Essential revisions:

1) It will be essential to evaluate the role of mechanosensitive ion channels in the repair mechanisms. Piezo1/2 and TRPV4 channels have been proposed to play a key role in many mechanobiological responses of different cell types, particularly the osteo-articular system. Therefore, their contribution should be evaluated by experiments in which their activity is assessed by loss-of-function e.g. implementing knockdown by applying si/shRNA and/or specific pharmacological approaches.

2) For the quantification of the western blots, total as well as phosphorylated forms of PKC and PKD should be used for normalization (phospho/total).

Other points to consider:

1) Some of the experiments were run in one type of cells and other ones in other cells? Is there any rationale on using one or the other cell type? If yes, it need to be explained clearly in the text.

2) Please define what the N numbers refer to in each figure.

3) For western blots, please provide the entire gel (as a supplementary figure).

4) Figures are too busy, particularly Figure 6. Can you please simply/reorganize this figure to facilitate its reading/understanding.

*Reviewer #1:*

"Mechanically-stimulated ATP release from murine bone cells is regulated by a balance of injury and repair" provides interesting and novel observations on mechanotransduction in osteoblasts that might be relevant for in-vivo function, with basic science and biomedical-translational relevance.

I would encourage the authors to explore more in-depth an eventual role of Piezo ion channels in the observed mechanotransduction. They go to great lengths to probe an array of molecules involved in mechanotransduction, Piezo should also be appreciated with more dedication.

Besides this issue, I have not identified additional major concerns.

*Reviewer #2:*

In this paper, Dr Komarova et al. describe a regulatory mechanism that controls the amount of ATP released in response to reversible membrane injury. They describe that exocytosis of ATP-containing vesicles can limit the efflux of intracellular ATP through the injured membranes (which is of a much larger scale). This is a newly described mechanism of cell resilience against mechanical forces.

I have a few questions that would like the authors to address:

1) You initially say that you tested osteoblasts from three different sources: two primary cultures (BM-OB and CB-OB) and one cell line (C2-OB). I may have missed it, but why did you choose to do some of the experiments in some of these cells and other ones in other cells? Is there any rationale on using one or the other cell type? If yes, it need to be explained clearly in the text.

2) In Figure 1 you show experiments in the 3 different cell types and there are significant differences between them. But these are not mentioned neither discussed in the manuscript. Could you please discuss.

3) Regarding n numbers; I understood that n=1 in the in vivo experiment means one animal. But in the in vitro experiments what do you define as n=1? Is one plate with x number of wells (biological replicates)? Or n=1 is one well? In this case n=x could mean that there are x wells coming all from the same original culture?

4) In the in vivo experiments, why did you only use female mice? Any rationale behind this? Could you explain in the text?

5) For the western blots I would like to see the entire gel, not just to show the band (if it makes the figure complicated you can add the gels as supplemental). But this is important for future replications.

6) Also for WB, why did you not measure total PKC and PKD. You should normalise the quantity of phosphoPKC or phosphoPKD to the total amount of these proteins (phsopho/total).

7) Figure 6 is far too busy and hard to understand. Could you please make it simpler or bigger? (Or divide in two figures when you include the bands and measurements of total kinases over phospho).

8) I think a schematic representation or diagram summarizing your results as a last figure could help a lot to understand the results and will give a clearer message.

*Reviewer #3:*

This paper adds significantly to a novel, recently proposed (Yu et al., 2017) hypothesis for explaining how bone adapts to mechanical stress. Physiologically relevant mechanical stresses are observed in vitro and in vivo to disrupt osteocyte plasma membranes. These membrane disruptions are not necessarily lethal but can be repaired allowing cell survival of this injury. However the disruption is thought to initiate signaling, such as a rise in intracellular calcium, that then leads to bone adaptation events. Here the authors show that ATP is released through osteocyte disruptions, a second signaling mechanism that might lead to an adaptive response. The sections of the paper dealing with repair are less strong as the authors do not have a technique for quantitative assay of this event.

---

## [Author Response]

Reviewer #1:[…] I would encourage the authors to explore more in-depth an eventual role of Piezo ion channels in the observed mechanotransduction. They go to great lengths to probe an array of molecules involved in mechanotransduction, Piezo should also be appreciated with more dedication.

Piezo channels have been implicated in mechanically-stimulated ATP release in red blood cells [1] and urothelial cells [2], and TRPV channels were shown to be involved in ATP release in urothelial [3, 4] and renal cells [5], cholangiocytes [6], as well as alveolar and bronchial epithelial cells [7]. Both Piezo 1/2 and TRPV4 channels have been previously demonstrated to be sensitive to Gd^3+^ [8, 9]. However, our data indicate that ATP release from mechanically-stimulated osteoblasts is not inhibited by Gd^3+^ (Figure 3—figure supplement 1B). We have additionally investigated the involvement of TRPV4 using its specific inhibitors HC 067047 [10] and GSK 2193874 [11], and are now providing new data addressing the involvement of Piezo1 using its specific inhibitor GsMTx-4 [12]. Consistent with the effects of Gd^3+^, specific inhibitors of Piezo 1 and TRPV4 were not effective in inhibiting mechanically stimulated ATP release from osteoblasts (Figure 3—figure supplement 1). Our findings are consistent with previous work that tested the involvement of Piezo1 and TRPV4 in osteoblasts using Gd^3+^ [13].

Besides this issue, I have not identified additional major concerns.Reviewer #2:[…] 1) You initially say that you tested osteoblasts from three different sources: two primary cultures (BM-OB and CB-OB) and one cell line (C2-OB). I may have missed it, but why did you choose to do some of the experiments in some of these cells and other ones in other cells? Is there any rationale on using one or the other cell type? If yes, it need to be explained clearly in the text.

During the study, we performed all the exploratory experiments in a C2C12-BMP2 cell line due to easy access and consideration for reducing animal use. To validate these findings, we used primary cultures of mouse osteoblasts. Since both bone-marrow-derived [14, 15] and compact-bone derived [16, 17] osteoblast cultures are used in the literature, we explored both cultures, and obtained qualitatively similar results, in spite of differences in the origin and differentiation state of the cells (pre-osteoblast-like in bone marrow derived cultures and late OB-osteocytes-like in compact bone-derived cultures). Since compact-bone derived late osteoblasts/osteocytes are closer to the in vivo model we used in this study, we continued using these cells for data validation throughout the study. Nevertheless, we included the data obtained in bone-marrow-derived cultures, since they demonstrate a side-by-side comparison for responses in osteoblastic cells obtained through different ex vivo protocols. While not a primary objective of this study, we believe that such data is important for understanding how different in vitro models behave. We have now included this clarification in the ‘Cell Cultures’ section of the Materials and methods.

2) In Figure 1 you show experiments in the 3 different cell types and there are significant differences between them. But these are not mentioned neither discussed in the manuscript. Could you please discuss.

As suggested, we have now discussed the differences in GAP junction involvement in bone-marrow-derived osteoblasts in the subsection “Mechanically-stimulated osteoblasts release ATP that induces calcium responses I n non-stimulated neighboring cells”.

3) Regarding n numbers; I understood that n=1 in the in vivo experiment means one animal. But in the in vitro experiments what do you define as n=1? Is one plate with x number of wells (biological replicates)? Or n=1 is one well? In this case n=x could mean that there are x wells coming all from the same original culture?

We have now clarified what n refers to in the ‘Statistical Analysis’ section of the Materials and methods, and in the legend for each figure.

4) In the in vivo experiments, why did you only use female mice? Any rationale behind this? Could you explain in the text?

C57Bl/6J male mice demonstrate significant in-cage fighting compared to females, which introduces higher variation in their background mechanical loading. This is expected to result in a less consistent mechanoresponsiveness and thus require more experimental sample size (i.e., more animals). As suggested, we have now clarified this in the ‘In-vivo Bone Loading’ section of the Materials and methods.

5) For the western blots I would like to see the entire gel, not just to show the band (if it makes the figure complicated you can add the gels as supplemental). But this is important for future replications.6) Also for WB, why did you not measure total PKC and PKD. You should normalise the quantity of phosphoPKC or phosphoPKD to the total amount of these proteins (phsopho/total).

As suggested in comments 5 and 6, we have now normalized phosphoPKC and phosphoPKD to total PKC or PKD (Figure 6E) and provided complete gels in Figure 6—figure supplement 4.

7) Figure 6 is far too busy and hard to understand. Could you please make it simpler or bigger? (Or divide in two figures when you include the bands and measurements of total kinases over phospho).8) I think a schematic representation or diagram summarizing your results as a last figure could help a lot to understand the results and will give a clearer message.

As suggested in comments 7 and 8, we have now moved some data from Figure 6 to Figure 6—figure supplement 2 and presented the schematic in Figure 7.

References:

1) Cinar, E., et al., Piezo1 regulates mechanotransductive release of ATP from human RBCs. Proc Natl Acad Sci U S A, 2015. 112(38): p. 11783-8.

2) Miyamoto, T., et al., Functional role for Piezo1 in stretch-evoked Ca(2)(+) influx and ATP release in urothelial cell cultures. J Biol Chem, 2014. 289(23): p. 16565-75.

3) Birder, L.A., et al., Altered urinary bladder function in mice lacking the vanilloid receptor TRPV1. Nat Neurosci, 2002. 5(9): p. 856-60.

4) Gevaert, T., et al., Deletion of the transient receptor potential cation channel TRPV4 impairs murine bladder voiding. J Clin Invest, 2007. 117(11): p. 3453-62.

5) Silva, G.B. and J.L. Garvin, TRPV4 mediates hypotonicity-induced ATP release by the thick ascending limb. Am J Physiol Renal Physiol, 2008. 295(4): p. F1090-5.

6) Gradilone, S.A., et al., Cholangiocyte cilia express TRPV4 and detect changes in luminal tonicity inducing bicarbonate secretion. Proc Natl Acad Sci U S A, 2007. 104(48): p. 19138-43.

7) Seminario-Vidal, L., et al., Rho signaling regulates pannexin 1-mediated ATP release from airway epithelia. J Biol Chem, 2011. 286(30): p. 26277-86.

8) Coste, B., et al., Piezo1 and Piezo2 are essential components of distinct mechanically-activated cation channels. Science (New York, N.Y.), 2010. 330(6000): p. 55-60.

9) Becker, D., et al., TRPV4 exhibits a functional role in cell-volume regulation. J Cell Sci, 2005. 118(Pt 11): p. 2435-40.

10) Everaerts, W., et al., Inhibition of the cation channel TRPV4 improves bladder function in mice and rats with cyclophosphamide-induced cystitis. Proc Natl Acad Sci U S A, 2010. 107(44): p. 19084-9.

11) Cheung, M., et al., Discovery of GSK2193874: An Orally Active, Potent, and Selective Blocker of Transient Receptor Potential Vanilloid 4. ACS Med Chem Lett, 2017. 8(5): p. 549-554.

12) Bae, C., F. Sachs, and P.A. Gottlieb, The mechanosensitive ion channel Piezo1 is inhibited by the peptide GsMTx4. Biochemistry, 2011. 50(29): p. 6295-300.

13) Genetos, D.C., et al., Fluid shear-induced ATP secretion mediates prostaglandin release in MC3T3-E1 osteoblasts. J Bone Miner Res, 2005. 20(1): p. 41-9.

14) Ode, A., et al., CD73/5'-ecto-nucleotidase acts as a regulatory factor in osteo-/chondrogenic differentiation of mechanically stimulated mesenchymal stromal cells. Eur Cell Mater, 2013. 25: p. 37-47.

15) Gharibi, B., et al., Adenosine receptor subtype expression and activation influence the differentiation of mesenchymal stem cells to osteoblasts and adipocytes. J Bone Miner Res, 2011. 26(9): p. 2112-24.

16) Brandao-Burch, A., et al., The P2X7 Receptor is an Important Regulator of Extracellular ATP Levels. Frontiers in Endocrinology, 2012. 3: p. 41.

17) Bakker, A.D. and J. Klein-Nulend, Osteoblast isolation from murine calvaria and long bones. Methods Mol Biol, 2012. 816: p. 19-29.